# Intranasal oxytocin interacts with testosterone reactivity to modulate parochial altruism
Boaz R. Cherki [1,2], Eyal Winter[2,3,4], David Mankuta[5], Shirli Zerbib[1] & Salomon Israel [1] ✉

The neuropeptide hormone oxytocin and the steroid hormone testosterone have received attention as modulators of behavior in the context of intergroup conflict. However, to date, their interactive effect has yet to be tested. Here, in a double-blind placebo-control design, 204 participants (102 female participants) self-administered oxytocin or placebo and completed an experimental economic game modeling intergroup conflict. Salivary testosterone ($n = 192$) was measured throughout the task to assess endogenous reactivity. As a caveat, even at this sample size, our derived power to detect small effects for 2- and 3-way interactions was relatively low. For male participants, changes in testosterone predicted willingness to sacrifice investments for the betterment of the group. Intranasal administration of oxytocin strongly diminished this effect. In female participants, we found no credible evidence for association between changes in testosterone and investments, rather, oxytocin effects were independent of testosterone. This 3-way interaction was of medium to large effect size (Odds Ratio 5.11). Behavior was also affected by social cues such as signaling of ingroup and outgroup members. Our findings provide insights as to the biological processes underpinning parochial altruism and suggest an additional path for the dual influence of oxytocin and testosterone on human social behavior.

As a social species, humans evolved to live in groups[1]. Competition between groups over limited resources (such as territory, money, status) often escalates into intergroup conflict[2,3]. Frequently, the rewards and losses associated with the outcomes of such conflicts are shared between all members of the group, regardless of their contribution to the group's success (i.e., public goods). While such a reward structure provides the greatest rewards to individual group members who do not contribute to the joint effort of the group (i.e., free riding)[4], it is well documented that during intergroup conflict, individuals are often willing to carry out self-costly actions to benefit their ingroup at the expense of rival groups[2,5,6].

Such acts of parochial altruism – the combination of ingroup favoritism and outgroup derogation – are thought to have emerged during human ancestry via the selective survival of groups, with lethal intergroup warfare playing a decisive role in this evolutionary selection process. At the individual level, both altruism towards ingroup members and hostility toward outgroup members are evolutionary disadvantageous, since both promote

costly, self-sacrificial behavior[7]. However, at the group level, groups composed mostly of altruists will outcompete groups whose members are mainly self-interested[8]. These phenomena are observed even under minimal group conditions where groups are constructed based on arbitrary criteria[2,9,10].

While computer simulation models, and ethnographic evidence have begun to elucidate the critical role that intergroup competition played in shaping the evolution of human social behavior[7,11], the psychobiological mechanisms underlying such behaviors remain poorly understood. Extant human research has largely focused on the independent effects of the neuropeptide hormone oxytocin[12,13] and on the steroid hormone testosterone[14,15], but the interactive effect of these hormones on intergroup behavior is not clear. Here, we use a laboratory-based task modeling intergroup conflict to examine the possibility that these two hormones may interact to regulate parochial altruism.

In the brain, oxytocin exerts varied effects on social behavior and cognition, either by its action as a neurotransmitter via projections from the

[1]Psychology Department, The Hebrew University of Jerusalem, Mount Scopus Campus, Mt. Scopus, Jerusalem 9190501, Israel. [2]The Federmann Center for the Study of Rationality, The Hebrew University of Jerusalem, Edmond Safra Campus, Givat Ram, Jerusalem 9190401, Israel. [3]Economics Department, The Hebrew University of Jerusalem, Mount Scopus Campus, Mt. Scopus, Jerusalem 9190501, Israel. [4]Management School, University of Lancaster, Lancaster LA1 4YX, UK. [5]Hadassah Medical Center, Department of Labor and Delivery, Kiryat Hadassah, Jerusalem 9112001, Israel. ✉e-mail: salomon.israel@mail.huji.ac.il

hypothalamus to limbic sites, or as a neurohormone via diffusion through the intracellular space to local or distant targets[16]. Findings from intranasal oxytocin studies in the context of intergroup relations are inconsistent. While several studies have demonstrated that intranasal oxytocin promotes participants' aggressive behavior toward the outgroup[17,18], particularly in competitive settings[12,19,20], other studies have shown that oxytocin promotes prosocial behavior toward the outgroup[21–23] and reduces intergroup aggression[13,24]. To date, these studies have been generally focused on male participants. However, across a range of social behaviors, oxytocin effects may differ by sex[25]. Thus, including female participants is essential in order to examine potential sex-dependent effects of oxytocin on behavior in intergroup conflict.

In addition to oxytocin, the steroid hormone testosterone has been examined as a modulator of behavior in the context of intergroup conflict. Besides its important role in the development of secondary sexual attributes, testosterone plays a key role in modulating human social behavior[26]. Testosterone has previously been associated with a variety of human antisocial behaviors, such as decreased generosity[27] and interpersonal trust[28], and increased aggression and violence[29]. To date, the few studies that have examined the role of testosterone in the context of intergroup conflict in humans have produced mixed findings[14,15,30,31]. Moreover, these studies, as well, excluded female participants, and focused almost exclusively on baseline testosterone levels. However, a recent meta-analysis examining the association between testosterone levels and aggressive behavior, a key element of parochial altruism, concluded that the association between baseline testosterone and aggressive behavior in humans is generally weak and inconsistent[32], suggesting instead that fluctuations in endogenous testosterone levels in response to social stimuli (henceforth, testosterone reactivity) may serve as a better indicator of the role of testosterone in modulating aggression.

The importance of testosterone reactivity in predicting behavior is consistent with the *challenge hypothesis*, which was initially developed to explain seasonal variations in testosterone concentration among monogamous male birds. The *challenge hypothesis* proposes that, in males, elevation in testosterone levels in response to a challenge in social status increases aggression[29,33]. The *challenge hypothesis* has received support from laboratory studies demonstrating a positive correlation between testosterone reactivity and aggressive behavior in male participants but not in female participants[32,34]. Thus, the first aim of our study is to examine sex-specific effects of testosterone reactivity on aggressive behavior in the context of intergroup conflict.

Interestingly, oxytocin and testosterone have opposing effects on a wide range of human social behaviors[35]. These opposing effects could be explained, at least in part, by the inhibitory effect of testosterone on oxytocin gene expression[36]. While animal models raise the intriguing possibility that oxytocin social effects may be contingent on testosterone levels[37], only a few studies have examined the interaction between oxytocin and testosterone in humans. In one of the few studies in humans examining these hormones together, high endogenous testosterone levels in female participants were associated with less attentional processing of infants' faces. This effect was canceled after intranasal oxytocin administration[38]. Another study demonstrated that testosterone reactivity is associated with the willingness to engage in competitive behavior (in male participants), but oxytocin administration cancels out this association[39]. Thus, a second aim of our study is to test whether oxytocin would moderate the association between testosterone reactivity and aggression towards the outgroup.

Finally, to date, studies examining the role of oxytocin or testosterone in modulating human behavior in the context of intergroup conflict have relied on paradigms in which decisions were made in the absence of information regarding the immediate behavior of others. However, in many real-life mixed-motive situations, individuals possess knowledge about intentions of others. Verbal signals such as communication, even if nonbinding (e.g., cheap-talk), induce changes in the behavior of ingroup and outgroup members. Specifically, laboratory studies demonstrate that when communication between participants is allowed, intragroup communication tends to increase aggression towards the rival group, while intergroup communication tends to decrease aggression between groups[40,41]. Animal models show that testosterone is associated with short-term signals of immediate behavioral intentions (e.g., puffing of the chest, threatening facial expressions, vocalizations)[42]; however, this association has yet to be tested in the context of intergroup games in humans. Thus, a third aim of this study is to test if testosterone reactivity is related to signaling of intentions, and whether oxytocin would moderate this association.

Towards these ends, 204 participants (102 female participants) self-administered oxytocin or placebo, and completed 30 rounds of the intergroup chicken game – a laboratory paradigm modeling the dynamics of intergroup conflict involving bilateral threats[43]. In this paradigm, participants are assigned to groups of two players, and each player decides whether to invest an endowment in the ingroup pool. If the number of investors in a group exceeds the number of investors in the rival group, its members receive a bonus. Investing the endowment is worthwhile only if it makes the difference between a tie to a victory, since in these cases, the expected received bonus is higher than the cost of the investment. In all other cases, however, rational players should keep the endowment to themselves.

## Methods
### Participants
Two hundred and four Israeli students (102 female participants; mean age = 24.47 (SD = 2.40)) participated in a double-blind, placebo-controlled, between-subject design experiment. Due to the difficulty of locating a relevant effect size for estimating the sample size, and the complexity of our design, the sample size was determined by power analysis that was conducted for unrelated experiment that participants completed in the same session[39]. This sample size is one of the largest reported in studies which applied oxytocin administration or measured testosterone reactivity in the context of aggressive behavior or intergroup dynamics (see Supplementary Note 1 for further details). This sample size allowed us to reliably detect three-way interactions of large effect sizes (OR > 6.71; see Supplementary Note 2, Supplementary Table 1, and Supplementary Fig. 1 for sensitivity analysis and further details). Participants were recruited in groups of eight or twelve, with an equal number of male and female participants in each of the 18 sessions between March 2017 and June 2018. Participants self-reported their biological sex (in Hebrew "min"), but information regarding gender (in Hebrew "migdar") was not obtained. Twelve participants (7 female participants) were excluded from the main analysis due to missing or unreliable saliva samples, leaving 192 participants (95 female participants) for further analysis. Participants were recruited across multiple campus sites to capture a broad assortment of undergraduate majors across the social science, humanities, life, and physical sciences. Before taking part in the experiment, participants self-reported they were < 35 years old, had no history of psychiatric or endocrine illness, smoked less than 15 cigarettes a day, and were not taking any prescription medications that might interact with oxytocin (e.g., antihistamines, Methylergonovine, blood pressure medications, amiodarone, particularly prophylactic vasopressors). For female participants, exclusion criteria also included current pregnancy or breastfeeding. Participants were instructed to refrain from smoking, eating, or drinking (except water) for 2 h before the experiment, and from physical activity, alcohol, and caffeine consumption for 24 h before the experiment. Participants received 100 New Israeli Shekels (NIS; ~ 25$) or equivalent course credit for completing the study, and an additional fee (ranging from 0 to 21 NIS) based on their performance and decisions. All participants signed a written informed consent form before they participated in the study. The study was conducted in accordance with the Declaration of Helsinki, and the protocol was approved by the Ethics Committee of Hadassah Medical Center (reference number: 0440-15-HMO). The study was not preregistered.

### Mood assessment
To test whether oxytocin had general effects on subjective state, participants completed a visual analog scale (VAS) questionnaire directly before

intranasal administration, and again at the conclusion of the experiment. The eight items assessed were working ability, tiredness, anxiety, anger, conversation ability, interpersonal closeness, concentration, and sadness. Each item was scaled from 1 ("not at all") to 10 ("very much"). Change scores between the first and second VAS assessment were not significantly affected by oxytocin (working ability: t(202) = 0.56, p = 0.575, Cohen's d = 0.08, 95% CI = [−0.32, 0.57]; tiredness: t(202) = 0.36, p = 0.720, Cohen's d = 0.05, 95% CI = [−0.48, 0.70]; anxiety: t(202) = 1.71, p = 0.089, Cohen's d = 0.24, 95% CI = [−0.06, 0.80]; anger: t(202) = 1.84, p = 0.068, Cohen's d = 0.26, 95% CI = [−0.04, 1.04]; conversation ability: t(202) = −0.32, p = 0.749, Cohen's d = −0.05, 95% CI = [−0.56, 0.40]; interpersonal closeness: t(202) = 1.80, p = 0.073, Cohen's d = 0.25, 95% CI = [−0.05, 1.09]; concentration: t(202) = 0.86, p = 0.390, Cohen's d = 0.12, 95% CI = [−0.30, 0.77]; sadness: t(202) = 1.20, p = 0.232, Cohen's d = 0.17, 95% CI = [−0.18, 0.73]).

## Saliva samples and testosterone assays

Testosterone levels were measured from saliva samples that were collected by passive drool into a small polystyrene tube at four time points throughout each session. Saliva samples were collected before oxytocin administration (Time-1), 25 m after administration (Time-2), right before the intergroup chicken game (Time-3—approximately 52 m after hormone administration), and right after the game (Time-4 – approximately 85 m after hormone administration). Saliva samples were frozen immediately following collection and stored at −80°C. At the end of the collection period, samples were assayed in our laboratory using a commercially available competitive enzyme immunoassay for testosterone (Salimetrics EIA, product number: 1-2402). All samples were run in duplicate, and the sample concentrations used in the analyses are the averages of the duplicates. Interassay coefficients of variation were 12.35% for low pools and 6.65% for high pools. The intrassay coefficient of variation was 5.76%. Samples for which the coefficient of variation exceeded 15% between duplicates, indicating unreliable assay results, were excluded from the analyses (overall 10 samples: Time-1—four samples, Time-2—one sample, Time-3—three samples, Time-4—two samples). The intrassay coefficient of variation for the remaining samples was 4.81%. Additionally, testosterone concentrations could not be obtained for 22 samples due to insufficient saliva provided during the collection periods (Time-1—six samples, Time-2—four samples, Time-3—four samples, Time-4—eight samples).

## Drug administration

Participants self-administered either 24 IU of oxytocin (three puffs of 4 IU in each nostril; Syntocinon spray; Novartis, Basel, Switzerland) or a placebo under the supervision of the experimenter. The placebo included all the Syntocinon ingredients except the active hormone. The administration of oxytocin or placebo was randomized within sex to ensure an equal number of male participants and female participants in every condition. Both the experimenter and the participants were blind to the drug condition, and participants could not differentiate between oxytocin and placebo (Fisher's exact test, p = 0.551).

## Intergroup chicken game

In the intergroup chicken game, participants are assigned to groups of two-player groups and play against a rival group. At the beginning of each round, each player receives an endowment of 2 MU, and decides, independently, whether to keep the endowment or to invest it in the ingroup pool. If the number of investors in a group exceeds the number of investors in the rival group, its members receive a bonus of 5 MU each; otherwise, no reward is given. Investments are not refunded, and players who do not invest their endowment get to keep it (see Fig. 1 for the payoffs-matrix).

One of the key features of the intergroup chicken game is that it allows players to signal their intentions (i.e., cheap-talk) during a 29 s period before their final decision. During this signaling period, players may change their signals as often as they want. The signal at the end of the 30th s is recorded as the final decision. Using real data, Supplementary Video 1 shows the

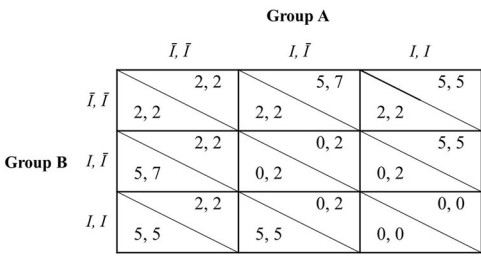

**Fig. 1 | Payoffs matrix of the intergroup chicken game.** The payoffs above the diagonals are for players in group A, whereas those below the diagonals are for the players of group B. The first payoff in each cell is for the left player of the group, and the second is for the right player. I represents 'invest', Ī represents 'not invest'.

perspective of a player during a single round of play, along with a visualization of the signals and the final decisions of their foursome (ingroup + rival group) in this round.

## Procedure

To control for diurnal rhythms in circulating oxytocin and testosterone levels, all experimental sessions were scheduled for 14:00, in keeping with the recommended guidelines for oxytocin administration studies[44]. After signing a written consent form, participants were seated in front of computers in cubicles, the first saliva sample (Time-1) was collected, and participants completed the mood assessment measure. Then, participants self-administered either oxytocin or placebo. Twenty-five minutes after the administration (in which participants were sitting in the lab with their cell phones turned off, instructed not to speak, and provided with National Geographic magazines to read), a second saliva sample (Time-2) was collected, and participants completed an unrelated experiment[39]. Approximately 52 m after hormone administration, the third saliva sample (Time-3) was collected, and participants were block-randomized (stratified by treatment) and assigned to two-player groups. Each group was composed of either two players who received oxytocin or two players who received placebo. Groups were composed of either two male participants, two female participants or one male and one female participant. Participants were not informed regarding the treatment and sex of their ingroup members and the outgroup members. Following the groups-assignment and the collection of the third saliva sample, approximately 55 m after hormone administration, participants completed 30 rounds of the intergroup chicken game against a competing group which received the opposite treatment. That is, a group that was composed of players who received oxytocin played against a group that was composed of players who received placebo (group-compositions remained constant throughout the experiment). Participants were not informed in advance the number of rounds to be played. Participants signaled their investment intentions by pressing (invest signaling) or releasing (non-invest signaling) the spacebar key. In each round, a brief reminder of the instructions was displayed at the center of the screen, as well as a countdown timer and four circles that represented the players' intention signaling (see Fig. 2).

When an ingroup member signaled 'invest', their circle was colored in green, and when an outgroup member signaled 'invest', their circle was colored in red. A summary of the decisions and payouts for the round was displayed on the screen at the conclusion of each round. This summary included the number of investors in the ingroup, the number of investors in the outgroup, player's payoff for this round, and player's cumulative payoffs.

Following the final round, participants completed a second mood assessment measure and demographic questionnaire and provided a fourth saliva sample (Time-4). At the end of the session, points were added up and cashed in at the rate of 1 Israeli Shekel for each 10 points. Following the experiment, participants were directed to another room and received payment privately. Participants were then briefed on the rationale and purpose of the study and dismissed individually (see Fig. 3 for the experiment's timeline).

## Statistical analyses

We conducted multilevel logistic and linear regression analyses with treatment (placebo/oxytocin), testosterone reactivity, and sex (female/male) as between-subjects variables. Alpha was set at 0.05 for all analyses.

The structure of our experimental design, in which rounds are nested within participants, participants are nested within dyads, and dyads are nested within foursomes, may lead to a violation of the nonindependence assumption. To account for potential dependency between observations, all models include participant ID as a random variable. In addition, we assessed dependence of observations at the dyad and the foursome level by computing the intraclass correlations (ICC). Although Kenny et al. suggested that at levels with two observations per cluster, an ICC lower than 0.45 allows referring to observations within clusters as independent, without increasing the chance for type-I error[45]; we took a conservative approach by adjusting standard errors by cluster. We corrected for clustering at the foursome level in order to allow for correlations among individuals at the most aggregate level[46]. We note that our findings are not contingent upon this analytical decision, as correcting for clustering at the dyad-level did not substantively affect the significance of the results (see Supplementary Table 2).

To account for known sex differences in testosterone levels (baseline levels in our sample; Male participants: $M = 150.87$, $SD = 55.11$, Female participants: $M = 50.96$, $SD = 19.63$, t-test on logarithmized values $(192) = -21.12$, $p < 0.001$, Cohen's $d = -3.03$, 95% CI of the logarithmized values difference = $[-1.20, -1.00]$, all testosterone values were standardized for each sex separately, by anchoring to the mean and the SD of testosterone concentrations at Time-1. Outliers were winsorized to $\pm 3$ SDs (Time-1—three sample, Time-2—four samples, Time-3—three samples, Time-4—one sample).

Testosterone reactivity from one time-point to another was assessed by regressing testosterone levels (standardized by sex) at the later time-point onto testosterone levels (standardized by sex) at the earlier time-point, and

saving the unstandardized residuals[47]. For example, testosterone reactivity from Time-3 (pregame) to Time-4 (post-game) was assessed by the unstandardized residuals of regressing testosterone levels at Time-4 onto testosterone levels at Time-3. Since the residuals represent changes in testosterone levels that are not explained by testosterone levels at the earlier time-point, this reactivity assessment is statistically independent of testosterone levels at the earlier point. We note that our findings are also not contingent upon this method of measuring testosterone reactivity, as measuring testosterone reactivity as ratio of testosterone levels between time-points, or absolute change in testosterone levels did not substantively affect the significance of the results (see Supplementary Note 3 and Supplementary Tables 3, 4).

To interpret null results, we performed the two one-sided test (TOST) procedure for equivalence testing. To so do, we first conducted sensitivity analyses to determine the smallest effect size that our models could detect with a statistical power of 0.8. We used Monte Carlo simulations for multiple effect sizes. For each effect size, we generated 1000 simulated datasets that were based on the characteristics of our sample (that is, the proportion of oxytocin and sex, and the mean and SD of testosterone reactivity), and on the parameters of the mixed logistic models that were conducted on the observed data. For each simulated dataset, we conducted the relevant regression model. We calculated the statistical power as the proportion of datasets, out of 1000, with odds ratios (OR) that differ significantly from 1. The smallest effect size that reached a statistical power of 0.8 was used as our equivalence bounds. We concluded that the test was statistically equivalent only if its 90% confidence intervals (CI), which represent an alpha level of 0.05, lied entirely within the equivalence bounds[48].

We used Stata (version 17.0)[49] to perform statistical analyses and generate Monte Carlo simulations, and R software (version 4.1.1)[50] to generate plots.

## Reporting summary

Further information on research design is available in the Nature Portfolio Reporting Summary linked to this article.

## Results

We conducted a series of multilevel logistic regressions to examine the factors affecting players' decisions whether to invest (or keep) their endowment across the 30 rounds of the intergroup chicken game. All analyses in the main text, unless otherwise specified, include participant ID as a random variable, and standard errors clustered by foursome (this denotes the members of both teams; for further details, see statistical analysis in the Method section).

### How do situational cues affect the likelihood to invest?

Before examining the biological factors that affect players' investments, we first provide a general description of behavior in the game. On average, participants invested in 61.69% (SD = 19.33%) of the rounds and earned 2.62 MU (SD = 2.12 MU) per round. Consistent with previous research[43], these outcomes show a stark deviation from economic models of rational behavior (see Supplementary Note 4 and Supplementary Fig. 2 for more

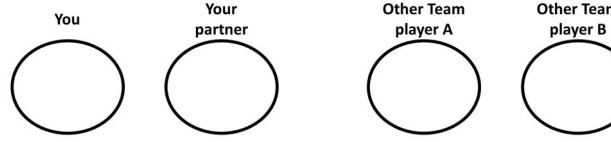

**Round number 1**

You received 2 points. You can choose to invest them in your group's pool (by pressing and holding the spacebar) or keep them to yourself (by releasing the spacebar).

Note that you may change your decision at any time, until the 30 seconds have expired.

Time remaining: 00:30

| You | Your partner | Other Team player A | Other Team player B |

**Fig. 2 | Example of screen layout during the intergroup chicken game.** Whenever ingroup member pressed the spacebar key (invest intention signaling) their circle was colored in green. Whenever an outgroup member pressed the spacebar key (invest intention signaling) their circle was colored in red.

**Fig. 3 | Experiment timeline.** Schematic timeline of the experimental session.

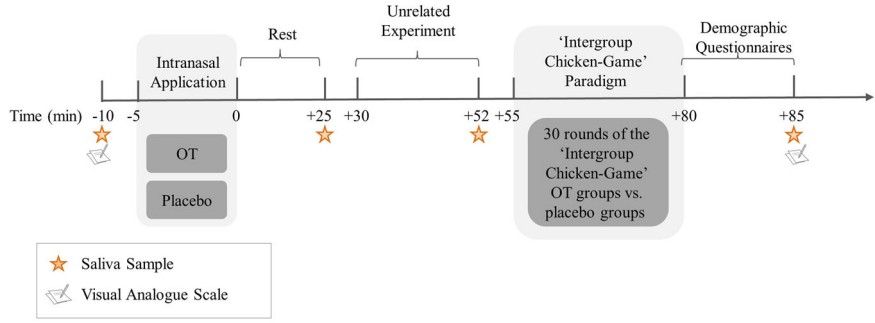

**Table 1 | Differences between the association of investment and signal at the 29th sec to signals at sec 1–28**

| Contrast | Odds ratio | P. value | 95% CI |
|---|---|---|---|
| sec 1 vs. sec 29 | 0.28 | < 0.001 | [0.18, 0.42] |
| sec 2 vs. sec 29 | 0.30 | < 0.001 | [0.20, 0.46] |
| sec 3 vs. sec 29 | 0.31 | < 0.001 | [0.21, 0.47] |
| sec 4 vs. sec 29 | 0.33 | < 0.001 | [0.23, 0.48] |
| sec 5 vs. sec 29 | 0.34 | < 0.001 | [0.24, 0.50] |
| sec 6 vs. sec 29 | 0.35 | < 0.001 | [0.25, 0.51] |
| sec 7 vs. sec 29 | 0.36 | < 0.001 | [0.24, 0.52] |
| sec 8 vs. sec 29 | 0.38 | < 0.001 | [0.25, 0.55] |
| sec 9 vs. sec 29 | 0.37 | < 0.001 | [0.25, 0.54] |
| sec 10 vs. sec 29 | 0.37 | < 0.001 | [0.26, 0.53] |
| sec 11 vs. sec 29 | 0.38 | < 0.001 | [0.27, 0.55] |
| sec 12 vs. sec 29 | 0.39 | < 0.001 | [0.27, 0.56] |
| sec 13 vs. sec 29 | 0.38 | < 0.001 | [0.26, 0.55] |
| sec 14 vs. sec 29 | 0.38 | < 0.001 | [0.26, 0.55] |
| sec 15 vs. sec 29 | 0.38 | < 0.001 | [0.26, 0.55] |
| sec 16 vs. sec 29 | 0.38 | < 0.001 | [0.26, 0.55] |
| sec 17 vs. sec 29 | 0.40 | < 0.001 | [0.28, 0.58] |
| sec 18 vs. sec 29 | 0.41 | < 0.001 | [0.28, 0.59] |
| sec 19 vs. sec 29 | 0.40 | < 0.001 | [0.28, 0.58] |
| sec 20 vs. sec 29 | 0.41 | < 0.001 | [0.29, 0.56] |
| sec 21 vs. sec 29 | 0.42 | < 0.001 | [0.30, 0.58] |
| sec 22 vs. sec 29 | 0.42 | < 0.001 | [0.31, 0.58] |
| sec 23 vs. sec 29 | 0.42 | < 0.001 | [0.31, 0.57] |
| sec 24 vs. sec 29 | 0.46 | < 0.001 | [0.34, 0.61] |
| sec 25 vs. sec 29 | 0.46 | < 0.001 | [0.35, 0.60] |
| sec 26 vs. sec 29 | 0.50 | < 0.001 | [0.40, 0.63] |
| sec 27 vs. sec 29 | 0.61 | < 0.001 | [0.51, 0.71] |
| sec 28 vs. sec 29 | 0.68 | < 0.001 | [0.59, 0.79] |

Post hoc contrasts of the differences between the association of signals at the 29th sec and players' likelihood to invest to the associations of each of the preceding 28 s of the signaling period and players' likelihood to invest. P values were Bonferroni corrected for 28 comparisons.

details). We also examined how situational cues that players encountered affected their decisions to invest (or keep) their endowments. These cues include the current signals from other players in the seconds leading up to the final decision for each round, and the previous history of investment outcomes by ingroup and outgroup members.

### Players' signals

We first examined the association between players' investments and their own signals to invest (or not). Rounds in which players ended up investing their endowment differed from rounds in which they did not in the duration of time in which players signaled 'invest'. Within the 29 s signaling period, players who ended up investing their endowment signaled 'invest' for 18.03 s (SD = 11.01 s), while players who did not invest their endowment signaled 'invest' for only 13.03 s (SD = 11.37 s; $b = 4.06$, SE = 0.44, $p < 0.001$, Cohen's $f^2 = 0.04$, 95% CI = [3.21, 4.92]). For each additional second that players signaled 'invest' the likelihood that they would actually invest was increased, on average, by 0.93% (OR = 1.05, SE = 0.005, $p < 0.001$, 95% CI = [1.04, 1.06]; See Supplementary Fig. 3-4 for the distribution of signals by second, treatment, and sex). There was also a first-mover advantage for signaling; the first player in the foursome who signaled 'invest' increased the chances that their team won the round, regardless of subsequent changes in the signal (OR = 1.55, SE = 0.28, $p = 0.013$, 95% CI = [1.10, 2.21]).

We next examined whether the effects of signaling were more pronounced in the seconds leading up to the final decision in each round. During each round of play, all four players signal their intentions (to invest or not); however, the final decision is only fixed at the end of the 30th s at each round. Thus, while such signals are not binding, they nevertheless represent the general intentions of the other players. The association between signaling 'invest' and actually investing strengthened with each passing second of the signaling period (OR = 1.02, SE = 0.003, p < 0.001, 95% CI = [1.02, 1.03]). Post hoc contrasts with Bonferroni correction show that the association between signals at the 29th s—the last second of the signaling period—with players' likelihood to invest was stronger compared to the associations between signals of each of the preceding 28 s (contrasts range = [0.38, 0.68], all $p$'s < 0.001; see Table 1); we therefore used players' signals at the 29th s to represent other players intentions at the time when final investment decisions were made.

Players' final investment decisions were influenced by the signaling behavior of their fellow ingroup member and the opposing outgroup. In cases where the ingroup member signaled 'invest' at the 29th s, players were themselves more likely to invest (OR = 3.48, SE = 0.43, $p < 0.001$, 95% CI = [2.74, 4.42]). Meanwhile, when opposing outgroup members signaled 'invest' at the 29th s, players were less likely to invest (OR = 0.50, SE = 0.04, $p < 0.001$, 95% CI = [0.43, 0.59]).

### How do prior ingroup and outgroup investment decisions affect the likelihood to invest?

To account for the investments of other players in previous rounds, we calculated a rolling average of investments from the first round to the previous round for the fellow ingroup member (ranging between 0 and 1) and for outgroup members (ranging between 0 and 2). A higher rate of previous investments of the ingroup member increased players' likelihood to invest (OR = 2.09, SE = 0.36, $p < .001$, 95% CI = [1.49, 2.93]). Previous investments of the outgroup members, however, were not significantly related to players' likelihood to invest (OR = 0.83, $SE = 0.14$, $p = .248$, 95% CI = [0.60, 1.14]).

### How do oxytocin, testosterone reactivity, and sex affect the likelihood to invest?

Next, we examined how investments were affected by oxytocin administration, testosterone reactivity during the intergroup chicken game, and sex. Testosterone reactivity was not dependent on oxytocin, time, or the oxytocin × time interaction (see Supplementary Note 5).

An analysis of the main effects for oxytocin, testosterone reactivity, and sex showed that while each factor on its own was significantly related to the number of seconds players signaled 'invest' (see Supplementary Note 6), there were no significant main effects of oxytocin (OR = 0.77, SE = 0.17, $p = 0.236$, 95% CI = [0.51, 1.18]), testosterone reactivity (OR = 1.07, SE = 0.19, $p = 0.700$, 95% CI = [0.76, 1.51]), or sex (OR = 1.06, SE = 0.15, $p = 0.679$, 95% CI = [0.81, 1.39]) on the final investment decision (see Table 2 Model 1), or on the association between 'invest' signals with the final decision (see Supplementary Note 6, Supplementary Fig. 5, and Supplementary Tables 5, 6 for additional signaling analysis). A model examining the two-way interactions between oxytocin, testosterone reactivity, and sex showed a significant oxytocin × sex interaction (OR = 1.79, SE = 0.46, $p = 0.025$, 95% CI = [1.08, 2.98]; see Table 2 Model 2). In male participants, the likelihood to invest the endowment did not differ between oxytocin or placebo (OR = 1.00, SE = 0.24, $p = 0.996$, 95% CI = [0.62, 1.62], equivalence bounds = [0.59, 1.69], 90% CI = [0.67, 1.50]). However, female participants under oxytocin were less likely to invest compared to female participants under placebo (OR = 0.56, SE = 0.13, $p = 0.015$, 95% CI = [0.35, 0.89]). Neither the oxytocin × testosterone reactivity (OR = 0.87, SE = 0.27, $p = 0.650$, 95% CI = [0.47, 1.60]) nor the sex × testosterone reactivity (OR = 1.04, SE = 0.35, $p = 0.904$, 95% CI = [0.54, 2.02]) interactions significantly predicted the probability to invest.

Most importantly, the three-way interaction between oxytocin, testosterone reactivity, and sex significantly predicted the likelihood of

**Table 2 | Multilevel logistic regression models of the players' likelihood to invest**

| Model | (1) | (2) | (3) | (4) | (5) |
|---|---|---|---|---|---|
| Oxytocin | 0.77 [0.51, 1.18] $p = 0.236$ | 0.58 [0.35, 0.94] $p = 0.026$ | 0.59 [0.37, 0.95] $p = 0.029$ | 0.63 [0.41, 0.96] $p = 0.034$ | 0.67 [0.43, 1.02] $p = 0.061$ |
| Male dummy | 1.06 [0.81, 1.39] $p = 0.679$ | 0.79 [0.51, 1.22] $p = 0.285$ | 0.80 [0.52, 1.23] $p = 0.311$ | 0.80 [0.54, 1.20] $p = 0.285$ | 0.81 [0.54, 1.20] $p = 0.291$ |
| Testosterone reactivity[a] | 1.07 [0.76, 1.51] $p = 0.700$ | 1.11 [0.63, 1.93] $p = 0.726$ | 0.80 [0.40, 1.61] $p = 0.535$ | 0.77 [0.41, 1.48] $p = 0.439$ | 0.74 [0.39, 1.42] $p = 0.363$ |
| Oxytocin × Male | | 1.79 [1.08, 2.98] $p = 0.025$ | 1.78 [1.09, 2.93] $p = 0.022$ | 1.85 [1.12, 3.06] $p = 0.016$ | 1.75 [1.03, 2.94] $p = 0.037$ |
| Oxytocin × Testosterone reactivity | | 0.87 [0.47, 1.60] $p = 0.650$ | 1.80 [0.72, 4.51] $p = 0.210$ | 1.98 [0.81, 4.85] $p = 0.136$ | 2.13 [0.89, 5.07] $p = 0.089$ |
| Male × Testosterone reactivity | | 1.04 [0.54, 2.02] $p = 0.904$ | 2.35 [0.92, 5.99] $p = 0.074$ | 2.49 [1.03, 6.00] $p = 0.043$ | 2.68 [1.12, 6.38] $p = 0.026$ |
| Oxytocin × Male × Testosterone reactivity | | | 0.20 [0.06, 0.65] $p = 0.007$ | 0.14 [0.04, 0.43] $p = 0.001$ | 0.12 [0.04, 0.39] $p < 0.001$ |
| Ingroup member's signal at the 29th s | | | | 3.62 [2.86, 4.58] $p < 0.001$ | 3.70 [2.89, 4.73] $p < 0.001$ |
| Outgroup members' signals at the 29th s | | | | 0.50 [0.43, 0.59] $p < 0.001$ | 0.50 [0.42, 0.59] $p < 0.001$ |
| Prior investments of ingroup member | | | | | 2.04 [1.39, 3.01] $p < 0.001$ |
| Prior investments of outgroup members | | | | | 1.02 [0.76, 1.38] $p = 0.872$ |
| Constant | 1.99 [1.50, 2.65] $p < 0.001$ | 2.30 [1.64, 3.22] $p < 0.001$ | 2.28 [1.63, 3.19] $p < 0.001$ | 2.43 [1.74, 3.41] $p < 0.001$ | 1.51 [0.91, 2.52] $p = 0.107$ |
| Log Pseudolikelihood | −3583.90 | −3581.59 | −3578.71 | −3296.38 | −3155.00 |
| AIC | 7177.79 | 7179.19 | 7175.43 | 6614.76 | 6335.995 |
| BIC | 7211.09 | 7232.46 | 7235.36 | 6688.01 | 6422.12 |
| Number of Participants | 192 | 192 | 192 | 192 | 192 |
| Number of Observations | 5760 | 5760 | 5760 | 5760 | 5568[b] |

Factors contributing to the investment decision per round, were assessed via a repeated multilevel mixed effect logistic regression model. Male dummy = 1 if participant is male, 0 otherwise. Values in each cell represent odds ratios. Brackets contain 95% confidence intervals. Standard errors were clustered by foursomes.
[a]Testosterone reactivity was assessed by regressing testosterone levels (standardized by sex) following the completion of the intergroup chicken game onto testosterone levels (standardized by sex) before the game was initiated and saving the unstandardized residuals.
[b]Model 5 does not include the first round since there are no prior investments.

investing (OR = 0.20, SE = 0.12, $p = 0.007$, 95% CI = [0.06, 0.65]; see Table 2 Model 3, Fig. 4, and Supplementary Fig. 6). In female participants, testosterone reactivity was not significantly associated with the decision to invest, neither under placebo (OR = 0.80, SE = 0.29, $p = 0.549$, 95% CI = [0.39, 1.64], equivalence bounds = [0.65, 1.53], 90% CI = [0.44, 1.46]), nor under oxytocin (OR = 1.44, SE = 0.38, $p = 0.171$, 95% CI = [0.86, 2.41], equivalence bounds = [0.65, 1.54], 90% CI = [0.93, 2.22]). In contrast, in male participants, the testosterone reactivity × oxytocin was a significant predictor of investments (OR = 0.35, SE = 0.13, $p = 0.006$, 95% CI = [0.17, 0.74]); while under placebo, testosterone reactivity was a significant predictor of the probability to invest (OR = 1.89, SE = 0.60, $p = 0.045$, 95% CI = [1.01, 3.53]), under oxytocin it was not (OR = 0.67, SE = 0.20, $p = 0.167$, 95% CI = [0.37, 1.18], equivalence bounds = [0.59, 1.69], 90% CI = [0.41, 1.08]). This three-way interaction was specific to testosterone reactivity, as pregame

testosterone levels were also not a significant predictor of investments either as a main effect (OR = 1.07, SE = 0.07, $p = 0.325$, 95% CI = [0.94, 1.21]), or in interaction with sex and oxytocin (OR = 0.73, SE = 0.20, $p = 0.258$, 95% CI = [0.43, 1.26]).

To examine the specificity of the three-way interaction (oxytocin × testosterone reactivity × sex) on the decision to invest, we tested whether these interactive effects could be accounted for by the situational cues (i.e., other players' signals, and investments at prior rounds). The effect of the oxytocin × testosterone reactivity × sex interaction on the likelihood to invest remained significant after controlling for other players' signals (OR = 0.14, SE = 0.08, $p = 0.001$, 95% CI = [0.04, 0.43]; see Table 2 Model 4) and for investments of other players in previous rounds (OR = 0.12, SE = 0.07, $p < 0.001$, 95% CI = [0.04, 0.39]; see Table 2 Model 5). An alternative situational factor for the investments of other players in previous rounds that could account for the effect of the oxytocin × testosterone reactivity × sex interaction on the likelihood to invest is round number. While round number, by itself, significantly predicted the likelihood to invest (OR = 1.01, SE = 0.004, $p = 0.009$, 95% CI = [1.003, 1.02]; see Supplementary Table 7), the three-way interaction on the likelihood to invest remained significant even after controlling for round number (OR = 0.20, SE = 0.12, $p = 0.007$, 95% CI = [0.06, 0.64]; see Supplementary Table 7).

Interestingly, while oxytocin, testosterone reactivity, and sex predicted investment decisions, they were not related to the payoffs participants received (ordinal mixed regression: OR = 0.62, SE = 0.34, $p = 0.374$, 95% CI = [0.21, 1.79], equivalence bounds = [0.24, 4.18], 90% CI = [0.25, 1.51]). This is because investment decisions can lead to either higher or lower payoffs, depending on context.

## Does the interaction between oxytocin and testosterone reactivity on investment persist across contexts?

The strong predictive effect of signals at the 29th s on the final decision (OR = 6.68, SE = 0.96, $p < 0.001$, 95% CI = [5.04, 8.85]) suggests that players could use these signals to intuit the final decision of the other players during the game. To further understand the nature of the oxytocin × testosterone reactivity interaction effect on the likelihood to invest in male participants, we examined this effect across different scenarios in the game. These scenarios were defined based on the signals of the other players in the foursome, in the last second (29th s) prior to the final decision.

In some scenarios, investments can be seen as monetary-driven, insofar as an investment results in increased earnings for the player. For example, when investing the endowment is likely to make the difference between a tie and a victory, given the signals of the other players at the last second before the final decisions. In these scenarios, the payoffs when investing the endowment (5 MU) are likely to be higher compared to keeping it (2 MU). However, in other scenarios, the motivation to invest may be driven by other considerations. For example, investing the endowment when it is not likely to make the difference between a tie to a victory (e.g., when the two members of the outgroup signal that they are going to invest) would result in lower payoffs (0 MU) than keeping it (2 MU). Under such scenarios, investments may be driven by other considerations such as status-seeking, and social dominance signaling.

We conducted exploratory analyses to examine the roles of testosterone and oxytocin on decisions in these two different contexts, we regressed the likelihood to invest on the oxytocin × testosterone reactivity interaction in male participants for each scenario. For scenarios in which investments were expected to increase payoffs, the oxytocin × testosterone reactivity interaction was not a significant predictor of the likelihood to invest (OR = 0.51, SE = 0.38, $p = 0.368$, 95% CI = [0.12, 2.19]; see Fig. 5a). In contrast, for scenarios in which investments were expected to decrease payoffs, the interaction between oxytocin to testosterone reactivity significantly predicted the likelihood to invest (OR = 0.33, SE = 0.12, $p = 0.002$, 95% CI = [0.16, 0.66]; see Fig. 5b). That is, while under placebo, testosterone reactivity was significantly associated with a greater number of nonmonetary-driven investments (OR = 2.06, SE = 0.63, $p = 0.018$, 95% CI = [1.13, 3.75]); under oxytocin it was not (OR = 0.68, SE = 0.20, $p = 0.187$,

**Fig. 4 | Likelihood to invest by oxytocin, testosterone reactivity, and sex.** Plots of the relationship between testosterone reactivity during the intergroup chicken game, treatment condition (oxytocin vs. placebo), and the likelihood of participants to invest their endowment. Plots are shown separately for female participants (**a**) and male participants (**b**). Testosterone reactivity is based on residuals of predicting testosterone levels (standardized by sex) after the intergroup chicken game by testosterone levels (standardized by sex) before the game. Shaded areas represent 95% confidence intervals. Panel a: n placebo = 1440 observations (48 participants), n oxytocin = 1410 (47 participants). Panel b: n placebo = 1410 observations (47 participants), n oxytocin = 1500 observations (50 participants).

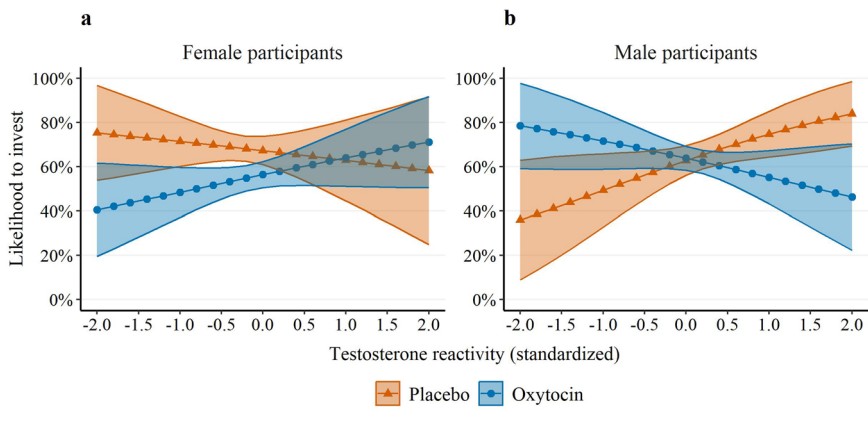

**Fig. 5 | Male participants' likelihood to invest by oxytocin, testosterone reactivity, and monetary considerations.** Plots of the relationship between testosterone reactivity during the intergroup chicken game, oxytocin, and the likelihood of male participants to invest their endowment. Plots are shown separately for monetary driven investments (**a**) and non-monetary driven investments (**b**). Testosterone reactivity is based on residuals of predicting testosterone levels (standardized by sex) after the intergroup chicken game by testosterone levels (standardized by sex) before the game. Shaded areas represent 95% confidence intervals. Panel a: n placebo = 374 observations (44 participants), n oxytocin = 347 (49 participants). Panel b: n placebo = 1036 observations (47 participants), n oxytocin = 1153 observations (50 participants).

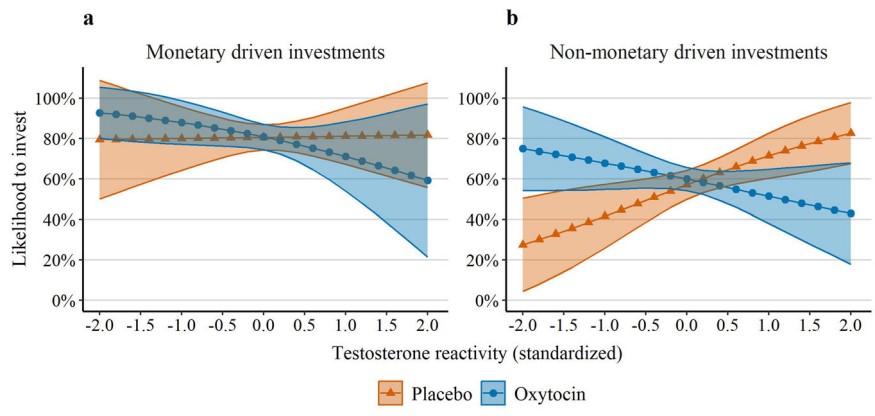

95% CI = [0.38, 1.21]). This interaction indicates that, in male participants, the general effect of the oxytocin × testosterone reactivity interaction on the likelihood to invest was driven, in part, by non-monetary considerations.

## Discussion

Current theories on the neurobiology of intergroup behavior postulate a central role for neuroendocrine systems. To date, despite an increased appreciation that neuroendocrine axes influence each other[51,52], research examining the role of oxytocin and testosterone on human intergroup behavior has largely focused on their independent effects[12,13,31]. In this study, we show that oxytocin and testosterone interact to shape competitive behavior within and between groups. In male participants, under placebo, rises in testosterone levels were associated with aggressive behavior towards the outgroup. Intranasal administration of oxytocin diminished this association. This oxytocin by testosterone interaction was specific to male participants, as in female participants, there was no credible evidence for an association between testosterone reactivity and outgroup aggression.

Previous research has linked testosterone reactivity to aggressive behavior in male participants in the context of competition between individuals[32,53,54]. Our findings extend this to show that testosterone reactivity increases aggression between competing groups as well. Interestingly, within the context of intergroup conflict, aggressive actions toward the outgroup take on a radically different purpose, as they weaken the relative strength of the outgroup and, therefore, serve to bolster the relative standing of the ingroup[55]. Our findings suggest that for male participants, testosterone reactivity indexes group status against competing outgroups. This is particularly evident when comparing monetary vs. non-monetary

investments. Greater testosterone reactivity predicted increased investments despite no immediate monetary benefit. Such behavior could be interpreted either as a signal to fellow ingroup members to spur investment, or as a threat to outgroup members to encourage them to shy away from investing. In either case, rather than promoting selfish behavior, testosterone reactivity is related to costly status-seeking behavior[56–58], which can be either pro or antisocial depending on context[59]. In female participants, however, we found no evidence for an association between testosterone reactivity and aggressive behavior neither under placebo nor under oxytocin. This finding is in line with research showing that the association between testosterone reactivity and competitiveness is more prominent in male participants than female participants[32,54,60]. Our findings suggest that in female participants, oxytocin acts independently to regulate behavior. Another possibility is that in female participants, oxytocin regulates the association between aggressive behavior and other sex hormones (estrogens, progesterone, follicle-stimulating hormone, luteinizing hormone), which were not assessed directly here.

Ongoing research demonstrates that the effects of oxytocin are not as simple as initially hypothesized. Effects of intranasal oxytocin are often sex-specific, context-dependent, and interact with other hormones[25,61]. Here we show that rather than affecting behavior directly, oxytocin's most pronounced effect in male participants was moderating the association between testosterone and behavior, an effect consistent with previous studies[38,39]. Evolutionary theories have suggested that in various living organisms, there is an interesting tradeoff between competitive behaviors and behaviors that enhance the welfare of others (e.g., parenting, cooperation)[62]. In male participants, oxytocin diminishes the association between testosterone reactivity and aggressive behavior, possibly by promoting tending behaviors such as parenting and romantic relations[63,64]. In female participants,

however, oxytocin directly decreased competitive behavior towards the outgroup. These findings are consistent with the idea that the biological mechanism underlying behavior during intergroup conflict may differ by sex. While in males, testosterone is responsible for regulating behavior, in females, it is oxytocin which regulates behaviors that are carried out to reduce risk for offspring[15,65]. While previous studies (in rodents) have shown that oxytocin may also modulate aggression in females, this role for oxytocin appears to be highly species and context specific. For example, while in female Syrian hamsters, oxytocin inhibits aggression[66], in female rats, the direction of the effect of oxytocin is modulated by trait anxiety[67]. Thus, our findings support an accumulating body of evidence showing that the effects of oxytocin may be conditioned on species, sex, trait background, and context[61,68].

The neurobiological mechanism underlying the moderating effect of oxytocin on the association between testosterone and behavior is not fully understood. However, this effect may be driven by the opposing effects of oxytocin and testosterone reactivity on brain regions that are involved in the expression of human aggression; the amygdala, which plays an important role in threat processing[69] and the orbitofrontal cortex (OFC), which is related to self-control and to the inhibition of impulsive behaviors[70]. While amygdala reactivity in the face of a threat is associated with higher inclination for aggressive responses[71–73]; in contrast, activity of the OFC, and its functional connectivity with the amygdala are related to the inhibition of impulsive aggression[71,74]. Previous studies have shown that rapid rises in testosterone levels enhance amygdala reactivity[75,76], decrease OFC reactivity[77], and suppress functional connectivity between these two regions[78,79]. Taken together, these findings have led to the idea that in the face of social threat, elevated testosterone levels increase the motivation for aggression by its effects on the amygdala, and suppress the inhibition of impulsive aggression by its effects on the OFC, and the amygdala-OFC connectivity[74]. Consistent with this idea, higher rates of investments, in our study, were associated with testosterone reactivity. Oxytocin administration, however, diminished the association between testosterone reactivity and investments. A finding consistent with previous studies showing that oxytocin decreases amygdala reactivity[80], increases OFC reactivity[81], and strengthens the connectivity between them[80], in response to social threat or competition.

Our study also sheds light as to the role of intention signaling in predicting and modulating aggressive behavior. While these effects were not affected by the oxytocin × testosterone reactivity interaction, they nevertheless provide novel insights as to the type of information players convey during an intergroup conflict, and how such signals influence the behavior of others. Players increased their investments in response to 'invest' signals of their fellow ingroup members and decreased their investment in response to such signals from the outgroup members. While this may seem trivial at first look, the reward structure of the intergroup chicken game induces players to establish themselves as investors to the outgroup, and simultaneously, as non-investors to their fellow ingroup member[82]. This structure also implies that players cannot commit themselves to any given action[82]. Nevertheless, our findings show that being the first player in a foursome to signal 'invest' increases the chances to win the round. Thus, despite being nonbinding, these intention signals were meaningful and positively correlated with participants' final decisions. Participants who ended up investing signaled 'invest' on average for 5 seconds more than participants who ended up not investing. This stands in contrast to game theoretical predictions, which argue that signals can be reliable only if they are difficult or impossible to cheat or too costly to bluff[83]. In contrast, our findings are in line with animal research, which have observed honest aggressive signals in several bird species, despite the ease of cheating or low cost of bluffing[84]. This is also consistent with evolutionary biology theories which suggest that honest system of aggressive signals can be evolutionary stable if the correlation between aggressive signals and behavior is positive but imperfect[85].

An alternative explanation is that such signaling is not intended to threaten outgroup members, but rather to encourage ingroup members to invest. While a clean separation of these motivations is not possible given our study design, the finding that the ingroup partner's history of investment in previous rounds predicted the player's current investment decisions, while outgroup history of behavior in previous rounds was not related to the player's decision suggests that ingroup coordination rather than outgroup derogation may also be a prominent motivation[86]. Future research will be needed to disentangle these competing mechanisms.

## Limitations

There are several limitations to our study that should be acknowledged. First, participants in our study completed the intergroup chicken game at 55 to 80 m after oxytocin administration. While there is no clear consensus regarding the putative window by which the effects of intranasal oxytocin on brain and behavior are the most prominent[87], some studies suggest that the most robust responses occur within a range of 45-70[88] minutes; raising one possible explanation regarding the insignificant main effect of oxytocin administration in male participants, due to waning oxytocin effects over time. In contrast, several other studies demonstrate more prolonged effects of oxytocin administration, suggesting that our behavioral window is within the time range of oxytocin effect[89–93]. To test for this possible issue, we conducted additional analyses in which we examined separately the behavioral results from 55-70 minutes and 70-80 minutes. The results are largely consistent across the two timeframes (see detailed results in Supplementary Table 8), suggesting that the extended experimental timeline did not lead to an attenuation of effects.

Second, although the sample used in this study included a mixed-sex sample and is one of the largest to date in the field of oxytocin administration and testosterone reactivity, given our sample size, we were well-powered to detect three-way interactions of large effect size, but only moderately powered to detect effects of medium effect size, and not adequately powered to detect three-way interactions of small effect sizes. Although our observed effect size is medium to large, initial discoveries often over-inflate effect sizes[94], and thus, future replications, with larger sample sizes and/or pre-registration, will be needed to verify our results.

Third, our study examined biological differences between males and females, and consequently, we report sex-based differences in neuroendocrine activity and behavior. However, social behaviors, such as intergroup conflict, are also the product of socially constructed roles and cultural context[95]. Future research would benefit by dissociating the roles of sex and gender in contributing to these processes.

## Conclusion

In conclusion, our findings extend the evidence base for the sex-specific effects of testosterone reactivity and oxytocin on human social behavior. Furthermore, these findings suggest an additional path for the dual influence of oxytocin and testosterone on human social behavior, by showing that within the context of intergroup conflict, the interaction between these hormones modulates aggressive behavior for the benefit of the ingroup. Consequently, our study provides insights into the neuroendocrine processes that underlie aggressive behavior in humans.

## Data availability

Access to the experimental data that support the findings of this study can be obtained through the OSF repository via this link.

## Code availability

Analysis code can be obtained through the OSF repository via the same link as the data.

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

## Acknowledgements

This research was supported by grants from the Israel Science Foundation (#1454/19), and from the German-Israeli Foundation for Scientific Research and Development (#I-2478-105.4/2017) to SI. The funders had no role in study design, data collection and analysis, decision to publish or preparation of the manuscript.

## Author contributions

B.R.C., E.W. and S.I. designed the research; D.M. gave medical support; S.Z. analyzed saliva samples; B.R.C. ran the experimental sessions; and B.R.C. and S.I. analyzed the data and wrote the paper.

## Competing interests

The authors declare no competing interests.
