## [Peer Review File · Communications Psychology]

2nd Mar 23

Dear Dr Israel,

Thank you for your patience during the peer-review process. Your manuscript titled "Intranasal Oxytocin Interacts with Testosterone Reactivity to Modulate Parochial Altruism" has now been seen by 3 reviewers, and I include their comments at the end of this message. They find your work of interest, but raised some important points. We are interested in the possibility of publishing your study in *Communications Psychology*, but would like to consider your responses to these concerns and assess a revised manuscript before we make a final decision on publication.

We therefore invite you to revise and resubmit your manuscript, along with a point-by-point response to the reviewers. Please highlight all changes in the manuscript text file. Please bear in mind that we would be reluctant to approach the reviewers again in the absence of revisions that comprehensively address all the concerns listed below.

Editorially, we consider it necessary that you address the following key concerns informed by the referee reports:

#1) Reviewers #1 and #2 raised the issue of the interpretation of non-significant results and that unless other approaches are used (e.g., Bayesian hypothesis testing, equivalence testing), the conclusions regarding non-significant results need to be updated. Non-significant results in NHST may be listed, but cannot support any claims or interpretation (i.e., non-significant p-values derived from traditional null hypothesis significance testing reflect either insensitivity or support for the null hypothesis). Please note that we have strict editorial requirements specifying reporting and interpretation of statistics, which you must adhere to as you address the referees' concerns.

In detail, all null results that are being interpreted must be supported by Bayesian statistics or equivalence testing (<https://journals.sagepub.com/doi/pdf/10.1177/2515245918770963>). For Bayesian statistics, we follow the convention detailed in Schönbrodt, F.D., Wagenmakers, E. Bayesian factor design analysis: Planning for compelling evidence. *Psychon Bull Rev* 25, 128–142 (2018) <https://rdcu.be/b6uOC> as a basis for the interpretation of BFs as evidence for the H1 or H0. Only evidence that is at least "moderate" may form the basis of interpretation.

Please also report full statistics for each individual test that is reported in the manuscript, including for non-significant results and additional analyses where "significance wasn't changed". In brief, you must report for each test the statistic(degrees of freedom) = value, p = value, effect size statistic = value, % Confidence Intervals = values, including for example on page 10.

#2) Reviewers #1 and #2 both raised the importance of an overall discussion of study limitations. We require that in revision, you perform the additional analyses requested by the reviewers to provide further support for your results, and transparently discuss the limitations that subsequently remain. Please include a section in limitations in the Discussion under its own subheading ("Limitations").

#3) All reviewers commented on sample size related issues, in terms of better transparency in reporting the sample size, improving the description of the sample size rationale, and implications for statistical power. The concern regarding the derived power to detect small effect sizes should be

mentioned as a caveat in the Abstract.

In addition, to comply with journal policy and guidelines (on which you will find more information via the links below), please also attend to the following issues:

#4) Please follow our SAGER-informed guidelines on studies reporting sex or gender differences, which you find in our editorial policies (summarized here: <https://www.nature.com/nature-portfolio/editorial-policies/ethics-and-biosecurity#research-on-human-populations>). Please state explicitly whether you are studying gender or sex differences. Studies of sex differences, should use the terminology: male participants/female participants. (Studies of gender differences would use men/women). Stylistically, “males” / “females” shouldn’t be used as nouns for human participants and the term "subjects" must be replaced by "participants". If your assessment of participant sex/gender didn’t dissociate between sex and gender, then this limitation must be in the discussion section (see also Sager guideline ii).

#5) Please revise your summary of the literature. In addition to the concerns raised by the referees, while the literature review may of course report animal studies, it must be made apparent for each reference whether it refers to findings in animals or humans. For example the sentence: “Studies show that T is associated with short-term signals of immediate behavioral intentions (e.g. puffing of the chest, threatening facial expressions, vocalizations) 40 , however this association has yet to be tested in the context of intergroup games.” appears to cite a paper studying song birds – in the context of a question regarding a behaviour studies in primates (intergroup games). In your literature summary, it would be best practice to highlight whether human studies assessed sex or gender (to the degree to which that is possible from the cited paper), and mention any uncertainties as a potential caveat.

Please use the following link to submit your revised manuscript, point-by-point response to the referees’ comments (which should be in a separate document to any cover letter) and the completed checklist:

[link redacted]

We hope to receive your revised paper within 4 months; please let us know if you aren’t able to submit it within this time so that we can discuss how best to proceed. If we don’t hear from you, and the revision process takes significantly longer, we may close your file. In this event, we will still be happy to reconsider your paper at a later date, provided it still presents a significant contribution to the literature at that stage.

We understand that due to the current global situation, the time required for revision may be longer than usual. We would appreciate it if you could keep us informed about an estimated timescale for resubmission, to facilitate our planning. Of course, if you are unable to estimate, we are happy to accommodate necessary extensions nevertheless.

Please do not hesitate to contact me if you have any questions or would like to discuss these revisions further. We look forward to seeing the revised manuscript and thank you for the opportunity to review your work.

Best regards,

Daniel Quintana

Daniel Quintana, PhD
Editorial Board Member
Communications Psychology
orcid.org/0000-0003-2876-0004

EDITORIAL POLICIES AND FORMATTING

Editorial Policy: Policy requirements (Download the link to your computer as a PDF.)

Furthermore, please align your manuscript with our format requirements, which are summarized on the following checklist:

Communications Psychology formatting checklist

and also in our style and formatting guide Communications Psychology formatting guide .

* **CODE AVAILABILITY:** All Communications Psychology manuscripts must include a section titled "Code Availability" at the end of the methods section. In the event of publication, we require that the custom analysis code supporting your conclusions is made available in a publicly accessible repository; at publication, we ask you to choose a repository that provides a DOI for the code; the link to the repository and the DOI will need to be included in the Code Availability statement. Publication as Supplementary Information will not suffice. We ask you to prepare code at this stage, to avoid delays later on in the process.

* **DATA AVAILABILITY:**

All Communications Psychology manuscripts must include a section titled "Data Availability" at the end of the Methods section or main text (if no Methods). More information on this policy, is

available at <http://www.nature.com/authors/policies/data/data-availability-statements-data-citations.pdf>.

At a minimum the Data availability statement must explain how the data can be obtained and whether there are any restrictions on data sharing. Communications Psychology strongly endorses open sharing of data. If you do make your data openly available, please include in the statement:

We recommend submitting the data to discipline-specific, community-recognized repositories, where possible and a list of recommended repositories is provided at <http://www.nature.com/sdata/policies/repositories>.

If a community resource is unavailable, data can be submitted to generalist repositories such as figshare or Dryad Digital Repository. Please provide a unique identifier for the data (for example a DOI or a permanent URL) in the data availability statement, if possible. If the repository does not provide identifiers, we encourage authors to supply the search terms that will return the data. For data that have been obtained from publicly available sources, please provide a URL and the specific data product name in the data availability statement. Data with a DOI should be further cited in the methods reference section.

REVIEWERS' EXPERTISE:

Reviewer #1 oxytocin interventions, cognition, economics games

Reviewer #2 oxytocin interventions, cognition

Reviewer #3 oxytocin interventions, cognition

REVIEWERS' COMMENTS:

Reviewer #1 (Remarks to the Author):

In this manuscript Cherki and colleagues investigate how externally administered oxytocin may interact with endogenous testosterone levels to affect behavior in intergroup conflict. In one of few studies investigating these effects on both males and females, the authors show that for males testosterone reactivity is associated with investments in an intergroup chicken game, when participants were incentivized to invest particularly in order to win the competing group. This effect seems to have been attenuated by oxytocin. In females on the other hand, they show that oxytocin reduced the likelihood of investing, but testosterone levels did not predict investing in the game. While I find the topic timely and interesting, and I applaud the authors' effort to investigate the interaction between the two hormones in both sexes I find there are some important issues that need to be addressed before recommending this manuscript for publication.

I hope the authors will find my comments helpful to improve their manuscript in this direction.

In reviewing the experimental timeline, it seems that the task of interest to the current paper (intergroup chicken game) took place at 55 to 80 minutes post OT administration. I would say that this is a rather delayed timeframe, as studies had shown that possibly the best window to assess the effects of intranasal OT administration is between 45 to 70 mins post administration. Especially in relation to the amygdala response, which the authors describe as a possible mechanism of action for the observed results, it has been clearly shown that 24 IU of OT elicited the most robust response within the 45-70 min window (see Spengler et al. 2017). Taking into account the relatively weak results in relation to OT, I would like to see the same analyses additionally performed after separating the trials from minutes 55 to 70 and from minutes 70 to 80. This might result in lower power but can be added as a supplementary analysis clarifying the potential effects of OT in the generally accepted time frames from a possible confound to the effects due to dropped OT levels because of the extended experimental timeline. Moreover, I believe it is essential to add this potential limitation to the discussion.

Spengler, F. B., Schultz, J., Scheele, D., Essel, M., Maier, W., Heinrichs, M., & Hurlmann, R. (2017). Kinetics and dose dependency of intranasal oxytocin effects on amygdala reactivity. *Biological Psychiatry*, 82(12), 885-894.

To their credit, the authors performed a thorough sensitivity analysis with respect to the power in their study to detect the observed effects. While the sample seems large enough, this analysis reveals that for most of the reported effects, the estimated statistical power was much less than what is recommended / needed (typically at least 80%), opening up the possibility that the probability of the reported results representing the true effects might be rather small. I would very much like to see this addressed in the discussion.

An overall discussion of limitations is missing, especially both for the points I outlined above and for the relative lack of results for females. It would add value to the manuscript if the authors discussed the observed and non-observed effects for females in the study as well. In the introduction, including females in this research is deemed essential, however, the results obtained for females (and again, even the lack of results) are not discussed. A general argumentation on the differential findings per sex would add to the paper and contribute to this indeed essential matter.

In the supplementary analyses the authors report that for males there was an increase in T concentration levels from T1 to T2. It would be useful if the authors could speculate as to why this was the case. Moreover, for the main analyses, the authors report that "all T values were standardized for each sex separately, by anchoring to the mean and the SD of T concentrations at Time-1" (pg. 20). I wonder if and how the results would change if for males T2 was taken as baseline for the task of interest.

I am curious as to why the authors selected to analyze the 30th second for the final decision. While it is intuitively clear that for pressing or lifting the finger from the space bar 1 second would be enough, I feel that 1 second for the actual decision might be slightly optimistic. Did the authors try the same analyses separating the first 28 seconds (for example) for signaling and keeping the 29th and 30th seconds for the final decision? I wonder how the results would look like in this case and suggest that if done so, the authors include a summary of results of this alternative analysis or alternatively a justification for the selection of the final second as the main dependent variable. It is

reported that the association between signals at sec 29 and likelihood to invest was stronger compared to sec 28. Could it be simply because participants did not have enough time to change their decision in the last second? In any case I believe it would be essential to include figures with the relevant descriptives, i.e., how many participants (per group/treatment) signaled/pressed at seconds 28,29,30? Similar plots for the course of all 30 seconds would be useful as well.

Another point related to the experimental timeline, is that the authors should report exactly what the participants did at rest (0-25 mins) and at the unrelated experiment (30-52 mins). Fatigue aside, it will be relevant to know whether these activities included any type of social interaction among participants or additional cognitive load as well as how the experimenters controlled for that if this is the case.

Subjects played 30 rounds with the same group. Though they did not know the number of trials in advance, one would expect there to be learning of the co-player and the outgroup behavior. Did trial number have any effect on investing or signaling? What about signaling history of the other participants? For example, one subject might notice through the course of rounds that Player X from the outgroup tends to signal early on for Y-Z seconds but not invest. While the authors report how previous investments affect (or not) decisions, it would be interesting to control for learning effects also by testing for past signaling patterns of the co-player and the outgroup.

Minor points

To improve readability in Figures 1 and 2, it would be better to remove some tick points from the axes (i.e., show only 20-point intervals instead of 10).

For a broad readership, the authors should provide a definition of T-reactivity when introduced.

Putting the findings and contributions of the paper in a larger theoretical ground could be essential to outline its contributions. How do the findings fit to ongoing theories for the role of oxytocin in human behavior?

Reading the manuscript further elicits the following minor questions that could be addressed:

Was T-reactivity greater or significantly different for males and females?

Mood assessment: what is the meaning of the variables “working capacity”, “conversation” and “closeness”? What were the questions used?

Were the groups (ingroup and outgroup) composed of both sexes? What did the participants know about the co-player and the other group? Both the composition of the groups and possible anonymity should be clearly reported.

Was signaling (cheap talk) affected by treatment? Did it differ between OT and PLC groups and/or in relation to T levels?

Reviewer #2 (Remarks to the Author):

Review for Intranasal oxytocin interacts with testosterone reactivity to modulate parochial altruism

Comments:

Abstract

- Is concise and well-written. However, in order to be more transparent about the sample size used, it would be useful to add the sample size here for readers to see up front.

Introduction

- Pg 3 – You state that findings are “equivocal” for OT and T on intergroup dynamics. The research is however very strong, particularly for OT, that there is a clear impact of these hormones on intergroup decision making. Indeed, the four references provided only provide supporting evidence. This either needs re-wording or new references added to support this point.
- Pg 4 – Relatedly, you provide a series of references to support the idea of mixed findings of OT. I would argue this is stretching the literature. Ref 13 can be argued to demonstrate that OT drives increased in-difference towards outgroups (see Daughters et al., 2017 for similar arguments – and also recruited a mixed-sex sample). Ref 21 is difficult to assess without data for individual investments but does illustrate that OT increased investment to the ingroup pool and that this increase in investment was reduced for global pools (where it should be noted ingroup members still received some benefit). Ref 22 is very context specific and to my knowledge not replicated. Ref 23 argues the opposite of “reduce[d] intergroup aggressions”, instead OT drove more premediated and coordinated aggression towards outgroup for greater profit. The literature is therefore far more complementary than the authors suggest. This should be clearly stated.
- Pg 6 – Although this is a personal choice, I feel there is too much detail on the methods for an ‘introduction’. A simpler holistic description of the game and aims would be sufficient, and moving the fuller description to the methods section.

Results

- The authors have done a good job of explaining a complicated set of results. However, one final full write out of the last result would be most welcome. Specifically, detailing participants behaviour in the OTxT-reactivity on likelihood to invest under OT conditions. I believe the difference between placebo and OT conditions here is a crucial finding of the paper and should be clearly laid out for readers.
- That being said, there are a lot of reported results here. Did the authors pre-register any of these analyses? Were any exploratory? This should be clearly stated.

Discussion

- I would imagine there are more references the authors could draw on to support their interpretations regarding T-reactivity driving increased investment despite not monetary gain.
- The authors rely on the lack of a significant finding under OT conditions to justify their interpretation that OT ‘cancels out’ T effects. My understanding is that only a Bayesian analysis would actually enable the authors to state this confidently. Otherwise, some caveating of their interpretation is required.
- The authors spend no time discussing the non-significant finding for females, or the use of mixed-sex sample or potential explanation for why this might have occurred.
- The authors present no limitations of their study.

Materials and Methods

- Some basic demographic information is missing about participants (mean age, SD age, ethnicity).

- It is important to be transparent about the sample sizes represented in the reported analyses. Although the authors provide a lot of information about their intended sample size and power calculations, ultimately, they had 192 participants, 96 in each drug condition, of which roughly 48 were male or female. Given the use of 3-way interactions, this final 'sample size' is perhaps the most relevant for conclusions drawn by the authors. This should be clearly stated.
- It would be useful for the authors to provide estimated or average time that saliva samples 3 & 4 were given. Specifically, how close they were to each other. In addition, if there was a time limit imposed on giving saliva samples, that should also be clearly stated.

Minor comments:

Pg. 4, last paragraph – testosterone is spelt out in full.

Reviewer #3 (Remarks to the Author):

Summary

The authors tested interactions between oxytocin (OT), testosterone (T), and sex on behavior in the context of intergroup conflict. They reported, using an experimental economic game modeling intergroup conflict, that for males, changes in endogenous T levels measured with saliva samples relate to the willingness of individuals to sacrifice investments for the betterment of the group. Intranasal administration of OT canceled out this relationship. In females, changes in endogenous T levels were unrelated to investments. Subjects' behavior was also affected by social cues such as the behavior and signaling of other ingroup and outgroup members, regardless of OT administration or T-reactivity.

General comments

The current study seems to be generally well-designed and the manuscript was well-written. The theme of study testing interaction between OT, T, and sex is timely and relevant. A few minor revision should be considered before publication.

Specific comments

In page 5, it is helpful for readers to describe the definition of T-reactivity at the first appearance.

In page 16 and supporting information, the rationale to set the sample size is still unclear. Specifically, although they said "Therefore, we decided to use a sample size of N = 204, which is one of the largest in the field." in the supporting information, the reason why the largest sample size is needed cannot be understood from the current manuscript.

In page 16 or results section, please clarify how to exclude the candidates with history of psychiatric or endocrine illness, how many candidates were tested eligibility, and the numbers of excluded subjects and the reasons to exclude.

In page 16, please describe the examples of medications that might interact with OT.

Response to Reviewers

Dear Editor,

Dear Reviewers,

We would like to thank you for your time and effort in reviewing our manuscript "Intranasal Oxytocin Interacts with Testosterone Reactivity to Modulate Parochial Altruism." We greatly appreciate your thoughtful and constructive feedback, which has enhanced the quality and rigor of our work.

In this letter, we provide a point-by-point response to each of the comments raised in the review, and detail the changes we made in the revised manuscript to address these concerns.

Sincerely,

Boaz Cherki and Salomon Israel

Reviewer #1:

In this manuscript Cherki and colleagues investigate how externally administered oxytocin may interact with endogenous testosterone levels to affect behavior in intergroup conflict. In one of few studies investigating these effects on both males and females, the authors show that for males testosterone reactivity is associated with investments in an intergroup chicken game, when participants were incentivized to invest particularly in order to win the competing group. This effect seems to have been attenuated by oxytocin. In females on the other hand, they show that oxytocin reduced the likelihood of investing, but testosterone levels did not predict investing in the game. While I find the topic timely and interesting, and I applaud the authors' effort to investigate the interaction between the two hormones in both sexes I find there are some important issues that need to be addressed before recommending this manuscript for publication.

I hope the authors will find my comments helpful to improve their manuscript in this direction.

Thank you. Your comments certainly helped us to improve our manuscript.

In reviewing the experimental timeline, it seems that the task of interest to the current paper (intergroup chicken game) took place at 55 to 80 minutes post OT administration. I would say that this is a rather delayed timeframe, as studies had shown that possibly the best window to assess the effects of intranasal OT administration is between 45 to 70 mins post administration. Especially in relation to the amygdala response, which the authors describe as a possible mechanism of action for the observed results, it has been clearly shown that 24 IU of OT elicited the most robust response within the 45-70 min window (see Spengler et al. 2017). Taking into account the relatively weak results in relation to OT, I would like to see the same analyses additionally performed after separating the trials from minutes 55 to 70 and from minutes 70 to 80. This might result in lower power but can be added as a supplementary analysis clarifying the potential effects of OT in the generally accepted time frames from a possible confound to the effects due to dropped OT levels because of the

extended experimental timeline. Moreover, I believe it is essential to add this potential limitation to the discussion.

Spengler, F. B., Schultz, J., Scheele, D., Essel, M., Maier, W., Heinrichs, M., & Hurlmann, R. (2017). Kinetics and dose dependency of intranasal oxytocin effects on amygdala reactivity. *Biological psychiatry*, 82(12), 885-894.

Thank you for this important comment. Indeed, the question of timing regarding putative OT effects following intranasal administration is a topic of growing interest. As noted in a recent review by Quintana et al. (2021), there is no consensus regarding the most efficacious timing of the effects of intranasal administered OT on brain activity or behavior. So, while as the reviewer notes, there is evidence for 45-70 minutes post administration to be the most efficacious window, there is also evidence for OT effects beyond this time frame. For example, one study (Striepens et al., 2013) demonstrated increased concentrations of OT in the CSF only 75 minutes following 24 IU intranasal administration of OT. In addition, a study which used MRI arterial spin labeling to detect changes in cerebral blood flow following intranasal OT (Paloyelis et al., 2016) observed increased activity in brain regions expected to express OT receptors beginning at 25 minutes and continuing up to 78 minutes (the latest time point assessed post-delivery). Another study found an effect of 40 UI intranasal OT administration on regional cerebral blood flow in several brain regions in a time period of 87-95 minutes following administration (Martins et al., 2020). Furthermore, another study showed effects of intranasal OT up to 102 min after administration on human brain function using resting-state EEG microstates (Zelenina et al., 2022). Finally, resting state amygdala–mPFC connectivity young female participants was increased 70-90 min post OT administration (Ebner et al., 2016).

Based on the reviewer's suggestion, we examined the possibility of waning OT effects by separating our analyses into earlier rounds which took place about 55-70 min following OT administration (rounds 1-20) and later rounds, which took place 70+ min following administration (rounds 21-30). A comparison of effect sizes between earlier and later rounds is presented in Table S5 (SI Appendix, p. 18). As can be seen in the Table, the main effect of OT was not stronger in the earlier rounds. Thus, it appears that the relatively late onset of the task did not result in a weaker

main effect of OT on behavior. Nevertheless, we refer to this possibility as a potential concern in the Limitation section in the Discussion (p. 17).

To their credit, the authors performed a thorough sensitivity analysis with respect to the power in their study to detect the observed effects. While the sample seems large enough, this analysis reveals that for most of the reported effects, the estimated statistical power was much less than what is recommended / needed (typically at least 80%), opening up the possibility that the probability of the reported results representing the true effects might be rather small. I would very much like to see this addressed in the discussion.

We would like to emphasize that the effect size of our main finding is not small, but rather medium to large. However, we now note in the Limitations section in the Discussion (p. 17) the relatively low power to detect small effects for 3-way interactions, and the issue of effect-size inflation for initial discoveries. We also mention this as a caveat in the Abstract (p. 2).

In addition, regarding statistical power for false negatives, as noted above, we now include equivalence tests for all null findings, and only interpret these findings when they are supported by equivalence tests.

An overall discussion of limitations is missing, especially both for the points I outlined above and for the relative lack of results for females. It would add value to the manuscript if the authors discussed the observed and non-observed effects for females in the study as well. In the introduction, including females in this research is deemed essential, however, the results obtained for females (and again, even the lack of results) are not discussed. A general argumentation on the differential findings per sex would add to the paper and contribute to this indeed essential matter.

We have now added a Limitations section to the Discussion (p. 17). In this section we refer to the late onset of the task relative to the OT administration, to issues of power, and to issues of sex vs. gender.

"There are several limitations to our study that should be acknowledged. First, participants in our study completed the intergroup chicken-game at 55 to 80 m after

OT administration. While there is no clear consensus regarding the putative window by which the effects of intranasal OT on brain and behavior are the most prominent (Quintana et al., 2021), some studies suggest that the most robust responses occur within a range of 45-70 (Spengler et al., 2017) minutes; raising one possible explanation regarding the insignificant main effect of OT administration in male participants, due to waning OT effects over time. In contrast, several other studies demonstrate more prolonged effects of OT administration, suggesting that our behavioral window is within the time range of OT effect (Ebner et al., 2016; Martins et al., 2020; Paloyelis et al., 2016; Striepens et al., 2013; Zelenina et al., 2022). To test for this possible confound, we conducted additional analyses in which we examined separately the behavioral results from 55-70 minutes and 70-80 minutes. The results are largely consistent across the two timeframes (see detailed results in Table S5 in the SI Appendix), suggesting that the extended experimental timeline did not lead to an attenuation of effects.

Second, although the sample used in this study included a mixed-sex sample and is one of the largest to date in the field of OT administration and T-reactivity, given our sample size we were well powered to detect three-way interactions of large effect size, but only moderately powered to detect effects of medium effect size, and not adequately powered to detect three-way interactions of small effect sizes. Although our observed effect size is medium to large, initial discoveries often over-inflate effect sizes (Ioannidis, 2008) and thus, future replications, with larger sample sizes and/or preregistration, will be needed to verify our results.

Third, our study examined biological differences between males and females, and consequently we report sex-based differences in neuroendocrine activity and behavior. However, social behaviors such as intergroup conflict, are also the product of socially constructed roles and cultural context (Wood & Eagly, 2012). Future research would benefit by dissociating the roles of sex and gender in contributing to these processes."

We also elaborate more in the Discussion regarding the results of female participants (that is, the direct effect of OT on female participants' behavior, and the lack of effect of T-reactivity):

"This finding is line with research showing that the association between T-reactivity and competitiveness is more prominent in male participants than female participants (Carré et al., 2013; Geniole et al., 2020; Geniole & Carré, 2018). Our findings suggest that in female participants, OT acts independently to regulate behavior. Another possibility is that in female participants OT regulates the association between aggressive behavior to other sex hormones (estrogens, progesterone, FSH, luteinizing hormone) which were not assessed directly here."

"...Previous research has suggested there is an interesting tradeoff between competitive behaviors and behaviors that enhance the welfare of others (e.g., parenting, cooperation; Del Giudice et al., 2015). In male participants, OT diminishes the association between T-reactivity and aggressive behavior possibly by promoting tending behaviors such as parenting and romantic relations (Rilling & Mascaró, 2017; Schneiderman et al., 2012). In female participants, however, OT directly decreased competitive behavior towards the outgroup. These findings are consistent with the idea that the biological mechanism underlying behavior during intergroup conflict may differ by sex. While in males, T is responsible for regulating behavior, in females, it is OT which regulates behaviors that are carried out to reduce risk for offspring (Muñoz-Reyes et al., 2020; Taylor et al., 2000). While previous studies (in rodents) have shown that OT may also increase aggression in females, this role for OT appears to be highly species and context specific. For example, while in female Syrian hamsters OT inhibits aggression (Harmon et al., 2002), in females rats the direction of the effect of OT is modulated by trait anxiety (de Jong et al., 2014). Thus, our findings support an accumulating body evidence that the effects of OT may be conditioned on species, sex, trait background, and context (Beery, 2015)."

In the supplementary analyses the authors report that for males there was an increase in T concentration levels from T1 to T2. It would be useful if the authors could speculate as to why this was the case. Moreover, for the main analyses, the authors report that “all T values were standardized for each sex separately, by anchoring to the mean and the SD of T concentrations at Time-1” (pg. 20). I wonder if and how the results would change if for males T2 was taken as baseline for the task of interest.

We speculate that although participants were not informed in advance about the nature of the tasks in the study, they could assume, due to the nature of other studies that are conducted in the lab, that the task will be based on an economic decision-making paradigm which may result in a challenge to status. Given that all experimental sessions were conducted with an even split of male and female participants, T levels for male participants may have also risen due to the presence of female participants in the lab. We are, however, hesitant to include this in the main manuscript, as it is rather speculative.

Regarding the main analyses, to clarify, T1 was not used as a baseline in our analysis. It was used only for anchoring the standardization by sex of T2-T4 (to allow for examination of change over time). Repeating the main analysis with T2 as the anchor for the standardization of T2-T4 in males does not substantively change the results (as seen in the Table below).

Model	(1)	(2)	(3)	(4)	(5)
OT	0.77 [0.50, 1.18] p = .236	0.58 [0.35, 0.94] p = .026	0.59 [0.37, 0.94] p = .027	0.63 [0.41, 0.96] p = .031	0.66 [0.43, 1.01] p = .056
Male dummy	1.06 [0.81, 1.38] p = .671	0.79 [0.51, 1.21] p = .286	0.80 [0.52, 1.23] p = .316	0.81 [0.54, 1.20] p = .291	0.81 [0.55, 1.20] p = .297
T-Reactivity	1.07 [0.77, 1.49] p = .690	1.12 [0.65, 1.94] p = .687	0.80 [0.40, 1.61] p = .537	0.78 [0.41, 1.48] p = .440	0.74 [0.39, 1.42] p = .363
OT × Male		1.79 [1.08, 2.97] p = .025	1.78 [1.09, 2.91] p = .022	1.84 [1.11, 3.03] p = .017	1.73 [1.03, 2.92] p = .038
OT × T-Reactivity		0.84 [0.47, 1.50] p = .563	1.80 [0.72, 4.50] p = .212	1.97 [0.81, 4.84] p = .137	2.13 [0.89, 5.06] p = .089
Male × T-Reactivity		1.04 [0.56, 1.96] p = .894	2.24 [0.91, 5.51] p = .079	2.37 [1.02, 5.52] p = .046	2.55 [1.11, 5.87] p = .028
OT × Male × T-Reactivity			0.21 [0.07, 0.68] p = .009	0.15 [0.05, 0.46] p = .001	0.14 [0.05, 0.41] p < .001
Ingroup member's signal at the 29th s				3.62 [2.86, 4.58] p < .001	3.70 [2.89, 4.73] p < .001
Outgroup members' signals at the 29th s				0.50 [0.43, 0.59] p < .001	0.50 [0.43, 0.59] p < .001
Prior investments of ingroup member					2.04 [1.39, 3.01] p < .001

Prior investments of outgroup members					1.02 [0.76, 1.38] p = .873
Constant	1.99 [1.50, 2.64] p < .001	2.30 [1.64, 3.22] p < .001	2.28 [1.63, 3.19] p < .001	2.44 [1.75, 3.41] p < .001	1.52 [0.92, 2.52] p = .105
Log Pseudolikelihood	-3583.89	-3581.54	-3578.71	-3296.37	-3154.98
AIC	7177.78	7179.08	7175.41	6614.74	6335.97
BIC	7211.075	7232.35	7235.34	6687.99	6422.09
Number of Participants	192	192	192	192	192
Number of Observations	5760	5760	5760	5760	5568

I am curious as to why the authors selected to analyze the 30th second for the final decision. While it is intuitively clear that for pressing or lifting the finger from the space bar 1 second would be enough, I feel that 1 second for the actual decision might be slightly optimistic. Did the authors try the same analyses separating the first 28 seconds (for example) for signaling and keeping the 29th and 30th seconds for the final decision? I wonder how the results would look like in this case and suggest that if done so, the authors include a summary of results of this alternative analysis or alternatively a justification for the selection of the final second as the main dependent variable. It is reported that the association between signals at sec 29 and likelihood to invest was stronger compared to sec 28. Could it be simply because participants did not have enough time to change their decision in the last second? In any case I believe it would be essential to include figures with the relevant descriptives, i.e., how many participants (per group/treatment) signaled/pressed at seconds 28,29,30? Similar plots for the course of all 30 seconds would be useful as well.

We apologize for the lack of clarity regarding the game details. The Intergroup Chicken-Game instructions state that the final decision is recorded at the very end of the 30th s of each round. As a countdown timer was shown on the participants' screen during each round, participants could prepare themselves to press or lift their finger several seconds before the time ran out. For participants who did not change their signal from the 29th s to the 30th s, the final decision would be equal to the signal at the 29th s, in any case. In contrast, for participants who did change the signal at the last second, there is no concern regarding a lack of time to change their signal. So, to

clarify, the decision regarding whether to invest or not is not based on an arbitrary cutoff of time, but rather is locked in at the end of the 30-second countdown. Once the countdown clock expires, the final decision is then presented to all 4 participants playing the game, where the calculation regarding points earned for each player is shown.

Regarding the question of the importance of the last second of signaling in predicting the final investment, the Figures below show the proportion of players who changed their signal by second, OT, and sex (first figure), and 'invest' signals by second, OT, and sex (second figure). As can be seen, many participants changed their decision at the last second (usually from 'invest' to 'not invest'). Thus, the issue does not seem to be a lack of activity in the last second, but rather a flurry of activity. These Tables are now presented in the SI appendix (Tables S4 and S5)

Another point related to the experimental timeline, is that the authors should report exactly what the participants did at rest (0-25 mins) and at the unrelated experiment (30-52 mins). Fatigue aside, it will be relevant to know whether these activities included any type of social interaction among participants or additional cognitive load as well as how the experimenters controlled for that if this is the case.

We now note that during the resting phase *"the participants were sitting in the lab with their cell phone turned off, instructed not to speak, and provided with National Geographic magazines to read"*. As to the unrelated experiment that the participants were engaging with prior to the chicken-game, all methods and results are available in the citation we have included (Cherki et al., 2021).

We do not think that cognitive load should be an issue here as participants were not instructed to remember anything throughout the experiment and were not told that they would be tested again following the experiment.

Subjects played 30 rounds with the same group. Though they did not know the number of trials in advance, one would expect there to be learning of the co-player and the outgroup behavior. Did trial number have any effect on investing or signaling? What about signaling history of the other participants? For example, one subject might notice through the course of rounds that Player X from the outgroup tends to signal early on for Y-Z seconds but not invest. While the authors report how previous investments affect (or not) decisions, it would be interesting to control for learning effects also by testing for past signaling patterns of the co-player and the outgroup.

We previously addressed the issue of the effects of previous rounds by including a rolling average of the number of investments by the partner and by the opposing group (p. 11). However, we now added supplementary analysis which directly examines the effect of round number on investments (p. 11, and Table S6 in the SI Appendix). As we noted in the Results section that *"An alternative situational factor for the investments of other players in previous rounds that could account for the effect of the OT × T-reactivity × sex interaction on the likelihood to invest is round number. While round number, by itself, significantly predicted the likelihood to invest*

(p = .009, see Table S6 in the SI Appendix), the three-way interaction on the likelihood to invest remained significant even after controlling for round number (p = .001, see Table S6 in the SI Appendix)".

We note here that round number was not a significant predictor of signaling (OR = 1.005, SE = 0.009, p = .550, 95% CI = [0.988, 1.023]).

While the broader question of analyzing the effect of players' signals at previous rounds on investments is an interesting idea, this is (unfortunately) beyond the scope of the paper. In each round, there are 2^{29} (= 536,870,912) options of signaling patterns, out of which participants used thousands of patterns. Thus, inputting the entire history of signaling would require a very different analytical strategy which is not the focus of the current manuscript.

Minor points

To improve readability in Figures 1 and 2, it would be better to remove some tick points from the axes (i.e., show only 20-point intervals instead of 10).

Thank you for this suggestion. The y-axis of Figures 1 and 2 now include tick marks only for 20-point intervals.

For a broad readership, the authors should provide a definition of T-reactivity when introduced.

Thank you for this suggestion. We have now added to the Introduction (p. 5) a short definition of T-reactivity within the context of the manuscript: "*...fluctuations in endogenous testosterone levels in response to social stimuli...*"

Putting the findings and contributions of the paper in a larger theoretical ground could be essential to outline its contributions. How do the findings fit to ongoing theories for the role of oxytocin in human behavior?

Thank you for this comment. We have taken this opportunity to expand our discussion to note how our findings relate to possible roles of T and OT in balancing tradeoffs between competitive and more nurturing behaviors, as well as sex-dependent

differences in the role of OT in regulating behavior, and how the effects of OT are often context and background specific (p.14-15).

Reading the manuscript further elicits the following minor questions that could be addressed:

Was T-reactivity greater or significantly different for males and females?

We have now added analysis of T-reactivity by sex (SI Appendix, p. 3). T-reactivity was greater for males than for females only from Time-1 to Time-2.

Mood assessment: what is the meaning of the variables “working capacity”, “conversation” and “closeness”? What were the questions used?

Participants were simply asked to rank their capacities on these items using a visual analog scale. Items were single-word descriptors of different categories of general functioning that either could affect behavior (e.g., working capacity) or have been hypothesized to be related to oxytocin and social behavior (e.g., closeness). Working capacity and conversational ability refer to the estimated ability of the participants to work and have a conversation at the time they filled out the questionnaire. Closeness refers to how much they felt interpersonal closeness at that time. We have now refined the terms in the manuscript (p. 19). The question was used to investigate whether the potential effects of OT on behavior were mediated by general effects on subjective state.

Were the groups (ingroup and outgroup) composed of both sexes? What did the participants know about the co-player and the other group? Both the composition of the groups and possible anonymity should be clearly reported.

Groups could be composed of either one male participant and one female participant, two male participants, or two female participants. Participants received no information regarding the composition of groups (who their partner or outgroup members were). Participants were also not informed about the sex and treatment of the other players in their foursome (ingroup and outgroup members; p. 22-23).

Was signaling (cheap talk) affected by treatment? Did it differ between OT and PLC groups and/or in relation to T levels?

We report these analyses in the Supporting Information (p. 4-5).

T-reactivity was positively related to the likelihood of players to signal 'invest' (OR = 2.12, SE = 0.20, $p < .001$, 95% CI = [1.76, 2.56]).

OT decreased the likelihood of players to signal 'invest' (OR = 0.76, SE = 0.07, $p = .001$, 95% CI = [0.60, 0.88]).

Reviewer #2:

Review for Intranasal oxytocin interacts with testosterone reactivity to modulate parochial altruism

Comments:

Abstract

•Is concise and well-written. However, in order to be more transparent about the sample size used, it would be useful to add the sample size here for readers to see up front.

We have now added to the abstract the sample size before the exclusion of participants with missing data of T-reactivity (n = 204) and after the exclusion (n = 192).

Introduction

•Pg 3 – You state that findings are “equivocal” for OT and T on intergroup dynamics. The research is however very strong, particularly for OT, that there is a clear impact of these hormones on intergroup decision making. Indeed, the four references provided only provide supporting evidence. This either needs rewording or new references added to support this point.

We removed the sentence regarding the equivocal finding. Instead, we state that up to date, no research has examined the interactive effect of these hormones on intergroup behavior.

•Pg 4 – Relatedly, you provide a series of references to support the idea of mixed findings of OT. I would argue this is stretching the literature. Ref 13 can be argued to demonstrate that OT drives increased in-difference towards outgroups (see Daughters et al., 2017 for similar arguments – and also recruited a mixed-sex sample). Ref 21 is difficult to assess without data for individual investments but does illustrate that OT increased investment to the ingroup pool and that this increase in investment was reduced for global pools (where it should be noted ingroup members still received some benefit). Ref 22 is very context specific and to my knowledge not replicated. Ref 23 argues the opposite of “reduce[d]

intergroup aggressions”, instead OT drove more premediated and coordinated aggression towards outgroup for greater profit. The literature is therefore far more complementary than the authors suggest. This should be clearly stated.

Note that with our revision to the comment above, the wording here is now specific to the question of whether OT promotes aggressive behavior toward outgroups. In the manuscript, we note that the findings here are ‘inconsistent.’ While the reviewer raises a number of caveats regarding our references suggesting that the findings here are more complimentary, we respectfully disagree with this assertion. While there are studies showing that OT increases aggression towards outgroups, there are also a number of studies showing that OT does not increase aggression towards outgroups, and in certain contexts reduces aggression / increases prosocial behavior.

Regarding the specific references the reviewer suggested as being more complimentary:

Ref 13: The authors of this paper state that "*oxytocin administration did not only reduce, it actually eliminated negative social behavior against out-group members*" (p. 124). Given a baseline (under placebo conditions) of reducing outgroup gains, indifference (under oxytocin) compared to negative social behavior is de facto being less aggressive.

Ref 21: results of this study show that as compared to placebo, under OT participants contributed more to the global pool which benefits both the ingroup and the outgroup. Note that investing in the global account here is inconsistent with a preference to benefit the ingroup only, or a preference to harm the outgroup/indifference to the outgroup. If participants were interested in benefiting the ingroup only then they should have contributed to the local pool. The finding that OT (vs. placebo) increases investment to the global pool is consistent with greater prosocial behavior towards outgroups (and ingroups).

Ref 23 (24 in the revised manuscript): While the reviewer notes the important finding regarding more premeditated and coordinated behavior towards outgroups, the authors of this study also state that "*Crucially, oxytocin increased the number of non-contributors in attacker groups but not in defender groups (Role × Treatment, $F(1, 78) = 5.043, p = 0.028, \eta^2 = 0.061, Figure 3B$)*". So it appears that the effects of OT on outgroup aggression are somewhat context specific. Given our objective here to

more clearly delineate findings which do not support OT-effects on increased outgroup aggression we have removed this reference.

This last reference notwithstanding, we believe that taken together with the "Oxytocin increases empathy to pain in the context of the Israeli–Palestinian conflict" (ref. 22), and "Oxytocin-enforced norm compliance reduces xenophobic outgroup rejection" (Marsh et al., 2017; ref. 23 in the revised manuscript), the findings suggest that in certain contexts OT may promote prosocial (or at least reduce antisocial behavior) toward the outgroup.

•Pg 6 – Although this is a personal choice, I feel there is too much detail on the methods for an ‘introduction’. A simpler holistic description of the game and aims would be sufficient, and moving the fuller description to the methods section.

The game description in the Introduction has been shortened, and the detailed description has been moved to the Methods section (p. 21).

Results

•The authors have done a good job of explaining a complicated set of results. However, one final full write out of the last result would be most welcome. Specifically, detailing participants behaviour in the OTxT-reactivity on likelihood to invest under OT conditions. I believe the difference between placebo and OT conditions here is a crucial finding of the paper and should be clearly laid out for readers.

We now have included in the SI Appendix Figure S6, which shows plots separated by treatment of the interaction between sex and T-reactivity on the likelihood to invest.

That being said, there are a lot of reported results here. Did the authors pre-register any of these analyses? Were any exploratory? This should be clearly stated.

We did not pre-register our analyses. We now note in the Results section on monetary-driven vs. strategic investments that “*We conducted exploratory analyses to examine the roles of T and OT on decisions in these two different contexts*”, and in the

SI appendix on signaling analyses that "*We conduct exploratory analyses to examine whether OT, T-reactivity, and sex, influenced players' signals during the signaling period*".

We also note in the Limitations section that "*future replications, with larger sample sizes and/or preregistration, will be needed to verify our results.*"

Discussion

•I would imagine there are more references the authors could draw on to support their interpretations regarding T-reactivity driving increased investment despite not monetary gain.

Thank you for this comment. We added to the Discussion (p. 13) references to Nave et al. (2018), Carré et al. (2010), and Geniole et al. (2017), which support the idea that increased T levels may increase status-driven behavior despite no monetary gain.

•The authors rely on the lack of a significant finding under OT conditions to justify their interpretation that OT ‘cancels out’ T effects. My understanding is that only a Bayesian analysis would actually enable the authors to state this confidently. Otherwise, some caveating of their interpretation is required.

Thank you for this important point. We have now included equivalence tests following null results, and have amended our wording accordingly. We performed equivalence testing using the two one-sided test (TOST) procedure. First, we conducted sensitivity analyses to determine the smallest effect size that our models could detect with a statistical power of 0.8. Due to the nested/hierarchical structure of our data (rounds are nested within participants, participants within dyads, and dyads nested within foursomes), the sensitivity analyses were based on Monte Carlo simulations at varying effect sizes. For each effect size, we generated 1000 simulated datasets that were based on the characteristics of our sample (that is, the proportion of OT and sex, and the mean and SD of T-reactivity), and on the parameters of the mixed logistic models that were conducted on the observed data. For each simulated dataset, we tested the relevant regression model. We calculated the statistical power as the proportion of datasets, out of 1000, with odds ratios that differ significantly from 1. The smallest effect size that reached a statistical power of 0.8 was used as our

equivalence bounds. We concluded that a test was statistically equivalent only if its 90% CI (which represent an alpha level of 0.05) lied entirely within its equivalence bounds (Quintana, 2018).

Specifically, in males, the relationship between T-reactivity to the likelihood to invest under OT was not equivalent and not different. That is, the 95% CI of this relationship did not include 1, and the 90% CI of this relationship was not entirely within the equivalence bounds (90% CI = [0.41, 1.08], equivalence bounds = [0.63, 1.58]). Accordingly, we replaced the words 'cancels out' with 'diminish'.

•The authors spend no time discussing the non-significant finding for females, or the use of mixed-sex sample or potential explanation for why this might have occurred.

We have now expanded our discussion in regards to the non-significant association between T-reactivity and the likelihood to invest in females, as well as the direct effect of OT on behavior in females in the Discussion section (p. 13). For more details, see response to Reviewer #1, above.

•The authors present no limitations of their study.

We have now added a Limitations section to the Discussion (p. 16). For greater detail, see our response to Reviewer #1, above.

Materials and Methods

•Some basic demographic information is missing about participants (mean age, SD age, ethnicity).

We have now added demographic information to the Participants section in the Methods (p. 17).

•It is important to be transparent about the sample sizes represented in the reported analyses. Although the authors provide a lot of information about their intended sample size and power calculations, ultimately, they had 192 participants, 96 in each drug condition, of which roughly 48 were male or

female. Given the use of 3-way interactions, this final ‘sample size’ is perhaps the most relevant for conclusions drawn by the authors. This should be clearly stated.

We have now clarified in the Abstract that T-reactivity was measured for 192 participants. We also add to the Abstract and to the Limitations section the relatively low statistical power to detect interaction effects.

•It would be useful for the authors to provide estimated or average time that saliva samples 3 & 4 were given. Specifically, how close they were to each other. In addition, if there was a time limit imposed on giving saliva samples, that should also be clearly stated.

We have now provided the estimated time of the collection of samples 3 (about 52 minutes after hormone administration) and 4 (about 85 minutes after hormone administration; p. 20, Saliva samples and T assays). We did not impose time limitation for the saliva collection, and none of the participants needed more than 5 minutes.

Minor comments:

Pg. 4, last paragraph – testosterone is spelt out in full.

Thanks. We changed it to "T".

Reviewer #3:

Summary

The authors tested interactions between oxytocin (OT), testosterone (T), and sex on behavior in the context of intergroup conflict. They reported, using an experimental economic game modeling intergroup conflict, that for males, changes in endogenous T levels measured with saliva samples relate to the willingness of individuals to sacrifice investments for the betterment of the group. Intranasal administration of OT canceled out this relationship. In females, changes in endogenous T levels were unrelated to investments. Subjects' behavior was also affected by social cues such as the behavior and signaling of other ingroup and outgroup members, regardless of OT administration or T-reactivity.

General comments

The current study seems to be generally well-designed and the manuscript was well-written. The theme of study testing interaction between OT, T, and sex is timely and relevant. A few minor revision should be considered before publication.

Thank you.

Specific comments

In page 5, it is helpful for readers to describe the definition of T-reactivity at the first appearance.

Thank you for this suggestion. We have now added to the Introduction (p. 5) a short definition of T-reactivity: "*...fluctuations in endogenous testosterone levels in response to social stimuli...*"

In page 16 and supporting information, the rationale to set the sample size is still unclear. Specifically, although they said "Therefore, we decided to use a sample size of N = 204, which is one of the largest in the field." in the supporting

information, the reason why the largest sample size is needed cannot be understood from the current manuscript.

The Sample Size section in the SI Appendix (p. 1) now includes a clear description of the sample size rationale. As we report in the manuscript, due to the difficulty locating relevant effect sizes for estimating the sample size, and the complexity of our design, we did not perform a priori power analysis for the current study. Instead, we used the same sample size that was determined by power analysis for an unrelated experiment that participants completed at the same sessions as the current study. Since this sample size would also be one of the largest samples for studies applying intranasal OT or measuring T-reactivity in the context of intergroup dynamics, we therefore decided to apply the same sample to the current study as well.

In page 16 or results section, please clarify how to exclude the candidates with history of psychiatric or endocrine illness, how many candidates were tested eligibility, and the numbers of excluded subjects and the reasons to exclude.

We now noted in the Participants section in the Methods (p. 18) that before taking part in the experiment, participants self-reported they had no history of psychiatric or endocrine illness.

However, we did not keep records of potential participants that were pre-excluded from the study.

In page 16, please describe the examples of medications that might interact with OT.

We have now added to the participants section in the Methods (p.17), antihistamines, Methylergonovine, amiodarone, and blood pressure medications, particularly prophylactic vasopressors as examples of medications that might interact with OT.

References

- Beery, A. K. (2015). Antisocial oxytocin: Complex effects on social behavior. *Current Opinion in Behavioral Sciences*, 6, 174–182.
<https://doi.org/10.1016/j.cobeha.2015.11.006>
- Carré, J. M., Campbell, J. A., Lozoya, E., Goetz, S. M., & Welker, K. M. (2013). Changes in testosterone mediate the effect of winning on subsequent aggressive behaviour. *Psychoneuroendocrinology*, 38(10), 2034–2041.
- Carré, J. M., Gilchrist, J. D., Morrissey, M. D., & McCormick, C. M. (2010). Motivational and situational factors and the relationship between testosterone dynamics and human aggression during competition. *Biological Psychology*, 84(2), 346–353.
- Cherki, B. R., Winter, E., Mankuta, D., & Israel, S. (2021). Intranasal oxytocin, testosterone reactivity, and human competitiveness. *Psychoneuroendocrinology*, 132, 105352.
<https://doi.org/10.1016/j.psyneuen.2021.105352>
- de Jong, T. R., Beiderbeck, D. I., & Neumann, I. D. (2014). Measuring virgin female aggression in the female intruder test (FIT): Effects of oxytocin, estrous cycle, and anxiety. *PloS One*, 9(3), e91701.
- Del Giudice, M., Gangestad, S. W., & Kaplan, H. S. (2015). *Life history theory and evolutionary psychology*.
- Ebner, N. C., Chen, H., Porges, E., Lin, T., Fischer, H., Feifel, D., & Cohen, R. A. (2016). Oxytocin's effect on resting-state functional connectivity varies by age and sex. *Psychoneuroendocrinology*, 69, 50–59.
- Geniole, S. N., Bird, B. M., McVittie, J. S., Purcell, R. B., Archer, J., & Carré, J. M. (2020). Is testosterone linked to human aggression? A meta-analytic

- examination of the relationship between baseline, dynamic, and manipulated testosterone on human aggression. *Hormones and Behavior*, *123*, 104644.
- Geniole, S. N., & Carré, J. M. (2018). Human social neuroendocrinology: Review of the rapid effects of testosterone. *Hormones and Behavior*, *104*, 192–205. <https://doi.org/10.1016/j.yhbeh.2018.06.001>
- Geniole, S. N., MacDonell, E. T., & McCormick, C. M. (2017). The Point Subtraction Aggression Paradigm as a laboratory tool for investigating the neuroendocrinology of aggression and competition. *Hormones and Behavior*, *92*, 103–116. <https://doi.org/10.1016/j.yhbeh.2016.04.006>
- Harmon, A. C., Huhman, K. L., Moore, T. O., & Albers, H. E. (2002). Oxytocin inhibits aggression in female Syrian hamsters. *Journal of Neuroendocrinology*, *14*(12), 963–969.
- Ioannidis, J. P. (2008). Why most discovered true associations are inflated. *Epidemiology*, 640–648.
- Marsh, N., Scheele, D., Feinstein, J. S., Gerhardt, H., Strang, S., Maier, W., & Hurlemann, R. (2017). Oxytocin-enforced norm compliance reduces xenophobic outgroup rejection. *Proceedings of the National Academy of Sciences*, *114*(35), 9314–9319. <https://doi.org/10.1073/pnas.1705853114>
- Martins, D. A., Mazibuko, N., Zelaya, F., Vasilakopoulou, S., Loveridge, J., Oates, A., Maltezos, S., Mehta, M., Wastling, S., & Howard, M. (2020). Effects of route of administration on oxytocin-induced changes in regional cerebral blood flow in humans. *Nature Communications*, *11*(1), 1160.
- Muñoz-Reyes, J. A., Polo, P., Valenzuela, N., Pavez, P., Ramírez-Herrera, O., Figueroa, O., Rodríguez-Sickert, C., Díaz, D., & Pita, M. (2020). The male warrior hypothesis: Testosterone-related cooperation and aggression in the

context of intergroup conflict. *Scientific Reports*, 10(1), 375.

<https://doi.org/10.1038/s41598-019-57259-0>

- Nave, G., Nadler, A., Dubois, D., Zava, D., Camerer, C., & Plassmann, H. (2018). Single-dose testosterone administration increases men's preference for status goods. *Nature Communications*, 9(1), 2433.
- Paloyelis, Y., Doyle, O. M., Zelaya, F. O., Maltezos, S., Williams, S. C., Fotopoulou, A., & Howard, M. A. (2016). A spatiotemporal profile of in vivo cerebral blood flow changes following intranasal oxytocin in humans. *Biological Psychiatry*, 79(8), 693–705.
- Quintana, D. S. (2018). Revisiting non-significant effects of intranasal oxytocin using equivalence testing. *Psychoneuroendocrinology*, 87, 127–130.
- Quintana, D. S., Lischke, A., Grace, S., Scheele, D., Ma, Y., & Becker, B. (2021). Advances in the field of intranasal oxytocin research: Lessons learned and future directions for clinical research. *Molecular Psychiatry*, 26(1), 80–91.
- Rilling, J. K., & Mascaró, J. S. (2017). The neurobiology of fatherhood. *Current Opinion in Psychology*, 15, 26–32.
- Schneiderman, I., Zagoory-Sharon, O., Leckman, J. F., & Feldman, R. (2012). Oxytocin during the initial stages of romantic attachment: Relations to couples' interactive reciprocity. *Psychoneuroendocrinology*, 37(8), 1277–1285.
- Spengler, F. B., Schultz, J., Scheele, D., Essel, M., Maier, W., Heinrichs, M., & Hurlemann, R. (2017). Kinetics and dose dependency of intranasal oxytocin effects on amygdala reactivity. *Biological Psychiatry*, 82(12), 885–894.
- Striepens, N., Kendrick, K. M., Hanking, V., Landgraf, R., Wüllner, U., Maier, W., & Hurlemann, R. (2013). Elevated cerebrospinal fluid and blood concentrations

of oxytocin following its intranasal administration in humans. *Scientific Reports*, 3(1), 1–5.

- Taylor, S. E., Klein, L. C., Lewis, B. P., Gruenewald, T. L., Gurung, R. A., & Updegraff, J. A. (2000). Biobehavioral Responses to Stress in Females: Tend-and-Befriend, Not Fight-or-Flight. *Psychological Review*, 107(3), 411–429.
- Wood, W., & Eagly, A. H. (2012). Biosocial construction of sex differences and similarities in behavior. In *Advances in experimental social psychology* (Vol. 46, pp. 55–123). Elsevier.
- Zelenina, M., Kosilo, M., Da Cruz, J., Antunes, M., Figueiredo, P., Mehta, M. A., & Prata, D. (2022). Temporal dynamics of intranasal oxytocin in human brain electrophysiology. *Cerebral Cortex*, 32(14), 3110–3126.

6th Oct 23

Dear Dr Israel,

Thank you for your patience during the peer-review process. Your manuscript titled "Intranasal Oxytocin Interacts with Testosterone Reactivity to Modulate Parochial Altruism" has now been seen by the same 3 reviewers as before, and I include their comments at the end of this message.

The reviewers find your work much improved and don't raise any further requests. However, there are some remaining editorial concerns. We are interested in the possibility of publishing your study in *Communications Psychology*, but would like to consider your responses to these concerns and assess a revised manuscript before we make a final decision on publication.

We therefore invite you to revise and resubmit your manuscript, along with a point-by-point response to the editorial concerns. Please highlight all changes in the manuscript text file.

In detail, we ask you to comprehensively address a set of concerns relating to statistical evidence and its reporting in the manuscript.

First, while equivalence testing is applied for some tests, this is not universally applied when making claims of "no effect" or "no difference". For example, "Previous investments of the outgroup members, however, were not related to players' likelihood to invest (OR = 0.83, SE = 0.14, $p = .248$, 95% CI = [0.60, 1.14])."

We cannot proceed with the manuscript unless all statements of no difference or no effect are supported by appropriate statistical evidence, which may be derived from equivalence testing or Bayesian statistics. Please note that reliable inferences on null results also require evidence that the study was suitably powered to detect the respective effects (see more comments on effect sizes below). Please adhere to our conventions on statistics reporting as detailed in the checklist and template linked below.

A second issue that we ask you to address is one of the justifications used for sample appropriateness in the 'participants' section. While the study is one of the largest reported in studies which applied oxytocin administration or measured testosterone reactivity in the context of aggressive behavior or intergroup dynamics, this simply indicates that the study design can reliably detect a wider range of effect sizes than previous studies, not that this range of effect sizes is necessarily important in and of itself. We ask you to report the effect sizes that can reliably be detected with the given study design. This is mentioned in the supplement (and in the limitations section as well), but should be mentioned in the 'participants' section of the main manuscript.

Finally, while we appreciate that the description of prior literature in the introduction is improved, we ask you to carefully review again how evident it is for each statement whether it was derived from animal or human studies. Please also note our conventions on the use of appropriate language in the context of gender and sex.

Male and female, which denote biological sex, can be used as nouns in reference to animals. Where you speak about the sex of human participants, please use male and female as adjectives (e.g., "male participants"). References to gender should ideally use the terms men/women etc. Please avoid jargon and refrain from using abbreviations for testosterone and oxytocin (e.g. "T") in the

main text.

Please note that your revised manuscript must comply with our formatting and reporting requirements, which are summarized on the following checklist:

Communications Psychology formatting checklist and also in our style and formatting guide Communications Psychology formatting guide .

Please use the following link to submit your revised manuscript, point-by-point response and the completed checklist:

[link redacted]

Please do not hesitate to contact me if you have any questions or would like to discuss these revisions further. We look forward to seeing the revised manuscript and thank you for the opportunity to review your work.

Best regards,

Marike, on behalf of
Daniel Quintana

Daniel Quintana, PhD
Editorial Board Member
Communications Psychology
orcid.org/0000-0003-2876-0004

EDITORIAL POLICIES AND FORMATTING

Editorial Policy: Policy requirements (Download the link to your computer as a PDF.)

* **CODE AVAILABILITY:** All Communications Psychology manuscripts must include a section titled "Code Availability" at the end of the methods section. In the event of publication, we require that the custom analysis code supporting your conclusions is made available in a publicly accessible repository; at publication, we ask you to choose a repository that provides a DOI for the code; the link to the repository and the DOI will need to be included in the Code Availability statement. Publication as Supplementary Information will not suffice. We ask you to prepare code at this stage, to avoid delays later on in the process.

* **DATA AVAILABILITY:**

All Communications Psychology manuscripts must include a section titled "Data Availability" at the end of the Methods section or main text (if no Methods). More information on this policy, is available at <http://www.nature.com/authors/policies/data/data-availability-statements-data-citations.pdf>.

At a minimum the Data availability statement must explain how the data can be obtained and whether there are any restrictions on data sharing. Communications Psychology strongly endorses open sharing of data. If you do make your data openly available, please include in the statement:

We recommend submitting the data to discipline-specific, community-recognized repositories, where possible and a list of recommended repositories is provided at <http://www.nature.com/sdata/policies/repositories>.

If a community resource is unavailable, data can be submitted to generalist repositories such as figshare or Dryad Digital Repository. Please provide a unique identifier for the data (for example a DOI or a permanent URL) in the data availability statement, if possible. If the repository does not provide identifiers, we encourage authors to supply the search terms that will return the data. For data that have been obtained from publicly available sources, please provide a URL and the specific data product name in the data availability statement. Data with a DOI should be further cited in the methods reference section.

REVIEWERS' COMMENTS:

Reviewer #1 (Remarks to the Author):

The authors have satisfactorily addressed all my comments. In my opinion, they also seem to have adequately addressed the other reviewers' comments and I am happy to recommend this manuscript for publication.

Reviewer #2 (Remarks to the Author):

The authors have addressed my concerns.

Reviewer #3 (Remarks to the Author):

The issues raised by the initial review were adequately addressed in the revision.

22nd Jan 24

Dear Dr Israel,

Your manuscript titled "Intranasal Oxytocin Interacts with Testosterone Reactivity to Modulate Parochial Altruism" has now been re-evaluated by Dr Daniel Quintana and myself. I am delighted to say that we are happy, in principle, to publish a suitably revised version in Communications Psychology under the open access CC BY license (Creative Commons Attribution v4.0 International License).

We therefore invite you to revise your paper one last time to address the remaining editorial requests. At the same time we ask that you edit your manuscript to comply with our format requirements and to maximise the accessibility and therefore the impact of your work.

EDITORIAL REQUESTS:

SUBMISSION INFORMATION:

OPEN ACCESS:

Communications Psychology is a fully open access journal. Articles are made freely accessible on publication under a CC BY license (Creative Commons Attribution 4.0 International License). This license allows maximum dissemination and re-use of open access materials and is preferred by many research funding bodies.

For further information about article processing charges, open access funding, and advice and support from Nature Research, please visit <https://www.nature.com/commspsychol/article-processing-charges>

At acceptance, you will be provided with instructions for completing this CC BY license on behalf of all authors. This grants us the necessary permissions to publish your paper. Additionally, you will be asked to declare that all required third party permissions have been obtained, and to provide billing information in order to pay the article-processing charge (APC).

* TRANSPARENT PEER REVIEW: Communications Psychology uses a transparent peer review system. On author request, confidential information and data can be removed from the published reviewer

reports and rebuttal letters prior to publication. If you are concerned about the release of confidential data, please let us know specifically what information you would like to have removed. Please note that we cannot incorporate redactions for any other reasons.

* CODE AVAILABILITY: All Communications Psychology manuscripts must include a section titled "Code Availability" at the end of the methods section. We require that the custom analysis code supporting your conclusions is made available in a publicly accessible repository at this stage; please choose a repository that generates a digital object identifier (DOI) for the code; the link to the repository and the DOI must be included in the Code Availability statement. Publication as Supplementary Information will not suffice.

* DATA AVAILABILITY:

[link redacted]

Best regards,

Marike

Marike Schiffer, PhD
Chief Editor
Communications Psychology

on behalf of

Daniel Quintana, PhD
Editorial Board Member
Communications Psychology